# A minus-end directed kinesin motor directs gravitropism in *Physcomitrella patens*

Yufan Li[1,6], Zhaoguo Deng[2,6], Yasuko Kamisugi [3], Zhiren Chen[2,4], Jiajun Wang[1], Xue Han[1], Yuxiao Wei[1,2], Hang He [1], William Terzaghi [5], David J. Cove[3], Andrew C. Cuming [3] & Haodong Chen [2,4✉]

Gravity is a critical environmental factor regulating directional growth and morphogenesis in plants, and gravitropism is the process by which plants perceive and respond to the gravity vector. The cytoskeleton is proposed to play important roles in gravitropism, but the underlying mechanisms are obscure. Here we use genetic screening in *Physcomitrella patens*, to identify a locus *GTRC*, that when mutated, reverses the direction of protonemal gravitropism. *GTRC* encodes a processive minus-end-directed KCHb kinesin, and its N-terminal, C-terminal and motor domains are all essential for transducing the gravity signal. Chimeric analysis between GTRC/KCHb and KCHa reveal a unique role for the N-terminus of GTRC in gravitropism. Further study shows that gravity-triggered normal asymmetric distribution of actin filaments in the tip of protonema is dependent on GTRC. Thus, our work identifies a microtubule-based cellular motor that determines the direction of plant gravitropism via mediating the asymmetric distribution of actin filaments.

[1] State Key Laboratory of Protein and Plant Gene Research, School of Advanced Agricultural Sciences and School of Life Sciences, Peking University, Beijing, China. [2] Center for Plant Biology, School of Life Sciences, Tsinghua University, Beijing, China. [3] Centre for Plant Sciences, University of Leeds, Leeds, UK. [4] Tsinghua-Peking Center for Life Sciences, Beijing, China. [5] Department of Biology, Wilkes University, Wilkes-Barre, PA, USA. [6] These authors contributed equally: Yufan Li, Zhaoguo Deng. ✉email: chenhaodong@tsinghua.edu.cn

Gravity is an ever-present environmental factor that regulates the growth direction and architecture of individual plants. In vascular plants, shoots grow upward (negative gravitropism) to harvest light and for gas exchange, while roots grow downward (positive gravitropism) to anchor in the soil and absorb nutrients and water[1]. Gravitropism has attracted scientific investigation for centuries. Charles Darwin demonstrated that specialized cells in seed plants could sense gravity and then transduce the signal to other cells to trigger bending, which he summarized in "*The Power of Movement in Plants*"[2]. Subsequently, the starch-statolith hypothesis and Cholodny–Went theory were put forward to explain gravity sensing and asymmetric growth, respectively[3,4]. Later, gravity-triggered asymmetric auxin distribution was shown to be regulated by polar auxin transport between cells by PIN proteins[5]. However, there is still a gap in understanding how the signal is transduced between the processes described by the two theories[1,6]. The microtubular cytoskeleton is an architectural feature of all eukaryotic cells and directly contacts many organelles that may play important roles in gravity signaling. In many models, the actin cytoskeleton is frequently assumed to be a component of gravitropism, but its role remains controversial since actin behaves as a positive regulator in some experiments but a negative regulator in others[7–10]. Additionally, there have been few studies about the role of microtubules in gravitropism[11,12].

By contrast, in tip-growing cells, gravity perception, signal transduction, and the gravitropic response take place within the same tip cell[13,14]. The protonemata of mosses grow by division and extension growth of the filament apical cell, and both microtubules and actin filaments are required for cell polarity during tip growth[15]. Microtubules were shown to play roles in the orientation of cell expansion, but the mechanisms were unclear[15,16]. Observations on the transient reversal of gravitropic growth by moss protonemal apical cells during mitosis implicate the possible involvement of microtubules in gravitropism[17,18]. Dynamic actin organization is necessary for tip cell expansion[19–21], and the position of an actin cluster near the tip cell apex dictates the growth direction[15]. The possible roles of actin microfilaments in gravitropism were assessed in dark-grown protonemata of the moss *Ceratodon*, but no major redistribution of the microfilaments was detected after gravistimulation[22].

Although many investigations have implied the involvement of the cytoskeleton in gravitropism, direct evidence is lacking. Here, we take advantage of *Physcomitrella patens* (*P. patens*) for a forward genetic screening to identify gravity signaling factors. *P. patens* produces large numbers of haploid protonemata that serve as good indicators of gravitropic responses, and possesses a well assembled and annotated genome[23]. We use several newly screened mutants, as well as a mutant *gtrC-5* (*gravitropism group C*) that was obtained via mutagenesis more than 30 years ago but with the corresponding gene not yet characterized[24,25], to identify the molecular nature of GTRC as a microtubule-based motor and to functionally analyze its roles in gravitropism.

## Results

### Mutagenesis screening for gravitropic mutants in *Physcomitrella patens*.
To take advantage of forward genetics in moss, we initiated a new screening in *P. patens* for gravitropic mutants (Fig. 1a). Protoplasts of *P. patens* (Gd ecotype) were generated for mutagenesis as described previously[26]. After resuspension in the PRMT medium, protoplasts were dispersed on the same medium containing agar in petri dishes overlaid with cellophane. The protoplasts were exposed to UV radiation and then incubated in darkness for 24 h[27]. Survivors were transferred to plates containing BCDAT medium, and then the plates were vertically

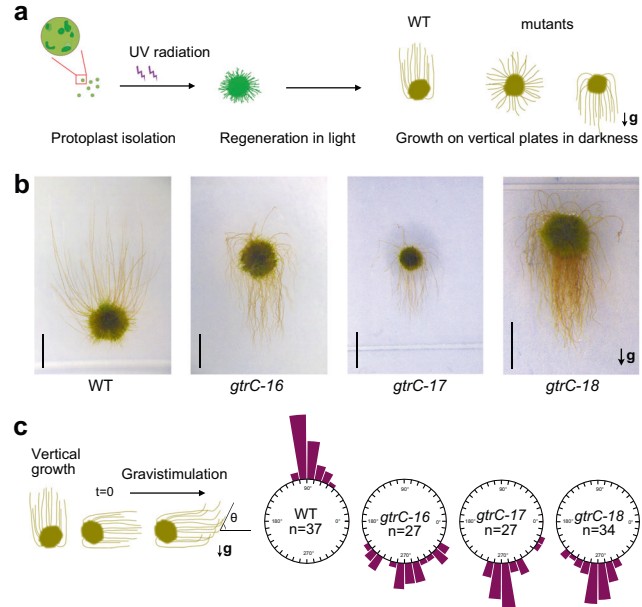

**Fig. 1 A group of mutants showing reversed gravitropic responses in *Physcomitrella patens*. a** Schematic diagram of the screening protocol for identifying gravitropic mutants in *P. patens*. Protoplasts of *P. patens* were isolated and treated with UV. Next, protonemata were regenerated in light and then grown on vertically oriented culture plates in the dark to record phenotypes. **b** Gravitropic phenotype of *P. patens* protonemata. Wild-type Gransden (Gd) strains of *P. patens* and UV-derived mutants were grown for around 3 weeks vertically in the dark. Scale bars, 3 mm. **c** Quantification of bending angles of protonemata growing in the dark for 3 weeks and then gravistimulated for 2 weeks. The schematic diagram shows how the angles were measured, and the bending angles are shown in circular histograms. Source data are provided as a Source Data file. In (**a–c**), arrows labeled with "g" indicate the directions of gravity vectors.

oriented in darkness for phenotypic screening (Fig. 1a). Several mutants whose protonemata grew wavy and downward were obtained, a clear reversal of growth polarity compared to wild type (Fig. 1b). The nomenclature of these mutants will be explained later after the loci are identified. Then, we did a further gravistimulation test by rotating the plates through 90 degrees, which further confirmed that these mutants all showed reversed gravitropic responses (Fig. 1c).

### Genetic mapping and confirmation of the *GTRC* locus.
*gtrC-5* was a mutant reported 35 years ago[24], in which gravity is sensed but the polarity of gravitropic growth is reversed. We crossed this mutant with a genetically divergent wild-type *P. patens* (Vx strain)[28] for genetic mapping. The mutant locus was mapped to an interval in Chromosome 2, and there is a nonsense mutation within locus *Pp3c2_9150* in this interval (Fig. 2a). For the newly obtained mutants, the first one isolated was crossed with wild-type *P. patens* (Vx strain), and the downward-growing segregants were combined and DNA was extracted for whole-genome sequencing. This mutation was also mapped to *Pp3c2_9150* (Fig. 2b). Other downward-growing mutants were checked for mutations within the same gene, and a number of additional allelic mutations within *Pp3c2_9150* were recorded. Consistent with the original nomenclature of *gtrC* mutants[17], these new mutants were named *gtrC-16, gtrC-17*, and *gtrC-18* (Fig. 1b, c). Correspondingly, the mapped gene was named *GTRC*, and the specific mutations are indicated in its gene and protein structures (Fig. 2c, d).

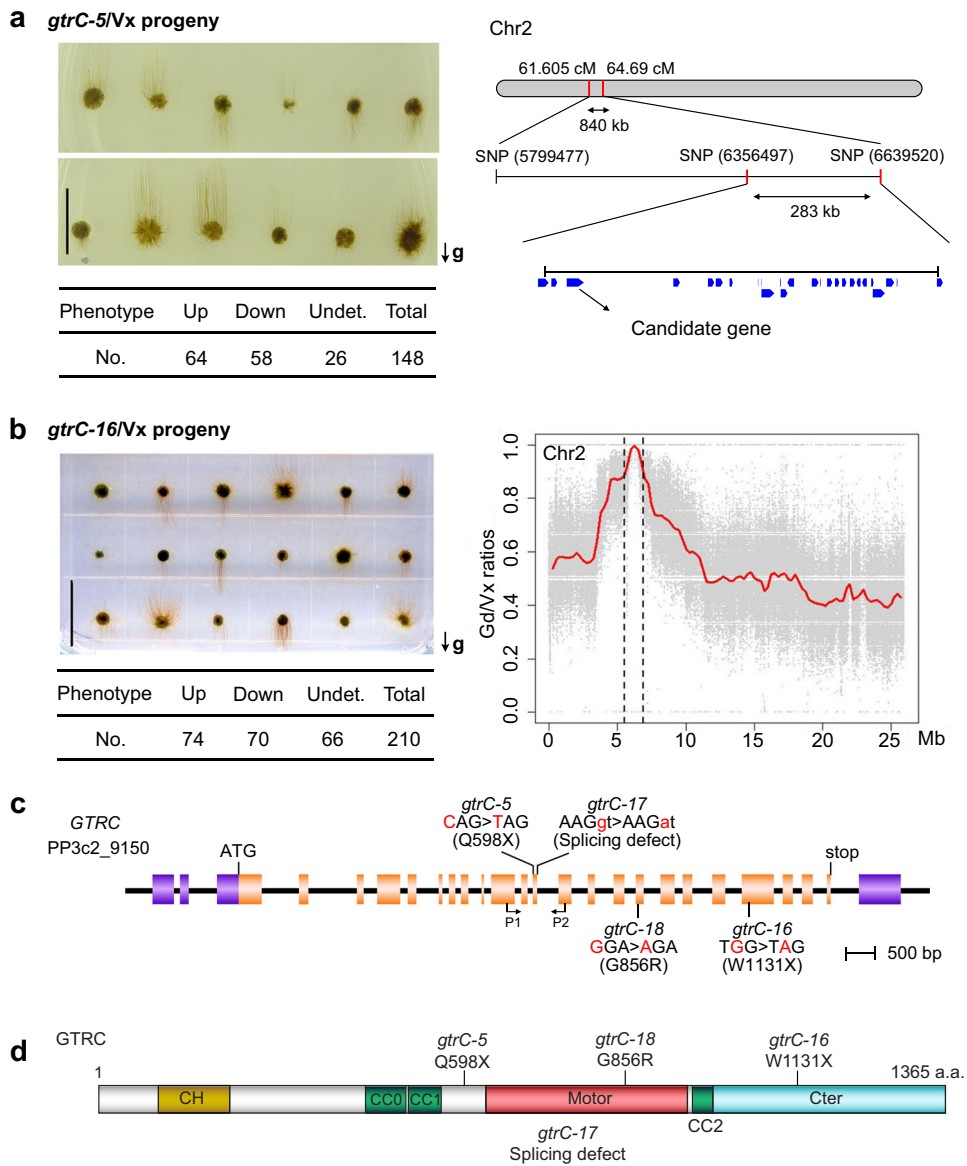

**Fig. 2 Genetic mapping of the *GTRC* locus. a** Mapping the *gtrC-5* mutant. Left, segregants from a *gtrC-5*/Vx hybrid spore capsule. Right: The genetic interval containing the *gtrC-5* locus was identified to a 840 kb region (GoldenGate: between 61.605 and 64.69 cM,) in the first round of mapping, and reduced to the interval of 283 kb distance in the second round of manual genotyping. A candidate gene with a nonsense mutation was identified in this reduced interval. Source data are provided as a Source Data file. **b** Mapping the *gtrC-16* mutants. Left, segregants from *gtrC-16*/Vx hybrids. Right, Gd/Vx SNP ratios in chromosome 2 derived from 48 *gtrC-16*/Vx hybrid progeny whose protonemata grew downwards. They were mixed and DNA was extracted for whole-genome sequence. Red lines are plots of mean values of Gd/Vx SNP ratios in a 0.5 Mb region with the sliding window set to 0.25 Mb. A candidate gene with nonsense mutation was identified. Source data are provided as a Source Data file. In **a**, **b**, the arrows with "g" show the direction of the gravity vector. The segregants were divided into three groups based on the growth direction of dark-grown protonemata: Up, Down, or Undetermined (Undet.). Scale bars, 10 mm. The numbers (No.) of each phenotype were shown in the tables below each image. **c** Gene structure and mutation sites of the *GTRC* locus. Purple box, untranslated region; orange box, coding exons. X indicates stop codon. **d** Protein structure and the mutation sites of GTRC. X indicates stop codon. Brown box, CH domain; green box, coiled coil region predicted by COILS program; red box, motor domain; blue box, C terminus of KCH proteins.

The gene *Pp3c2_9150* encodes a motor-domain protein PpKCHb/Kinesin 14-IIb[29,30] (Fig. 2d). Targeted mutagenesis of the *Pp3c2_9150* locus in wild type to produce the mutation present in *gtrC-5* was carried out. Four additional (silent) bases were altered in order to facilitate discrimination between WT and mutant sequences by allele-specific PCR (Fig. 3a). Those lines successfully converting the amino acid $Q^{598}$ to a stop codon resulted in a downward-growing *gtrC* phenotype, supporting the identification of this gene as a determinant of the gravitropic response (Fig. 3b–d). Further independent mutant alleles GTRC-

HRKO and GTRC-Cas9KO were generated in the wild-type background via homologous recombination and CRISPR strategies respectively, and both showed downward growth phenotypes similar to the previously identified *gtrC* mutants (Fig. 3e and Supplementary Fig. 1a). These results demonstrate the conclusion that loss of *GTRC* function leads to the observed reversed gravitropic phenotype.

Furthermore, the mutated *GTRC* sequence in *gtrC-16* was replaced with either wild-type genomic DNA or cDNA, and both restored the normal gravitropic phenotype (Fig. 3f and

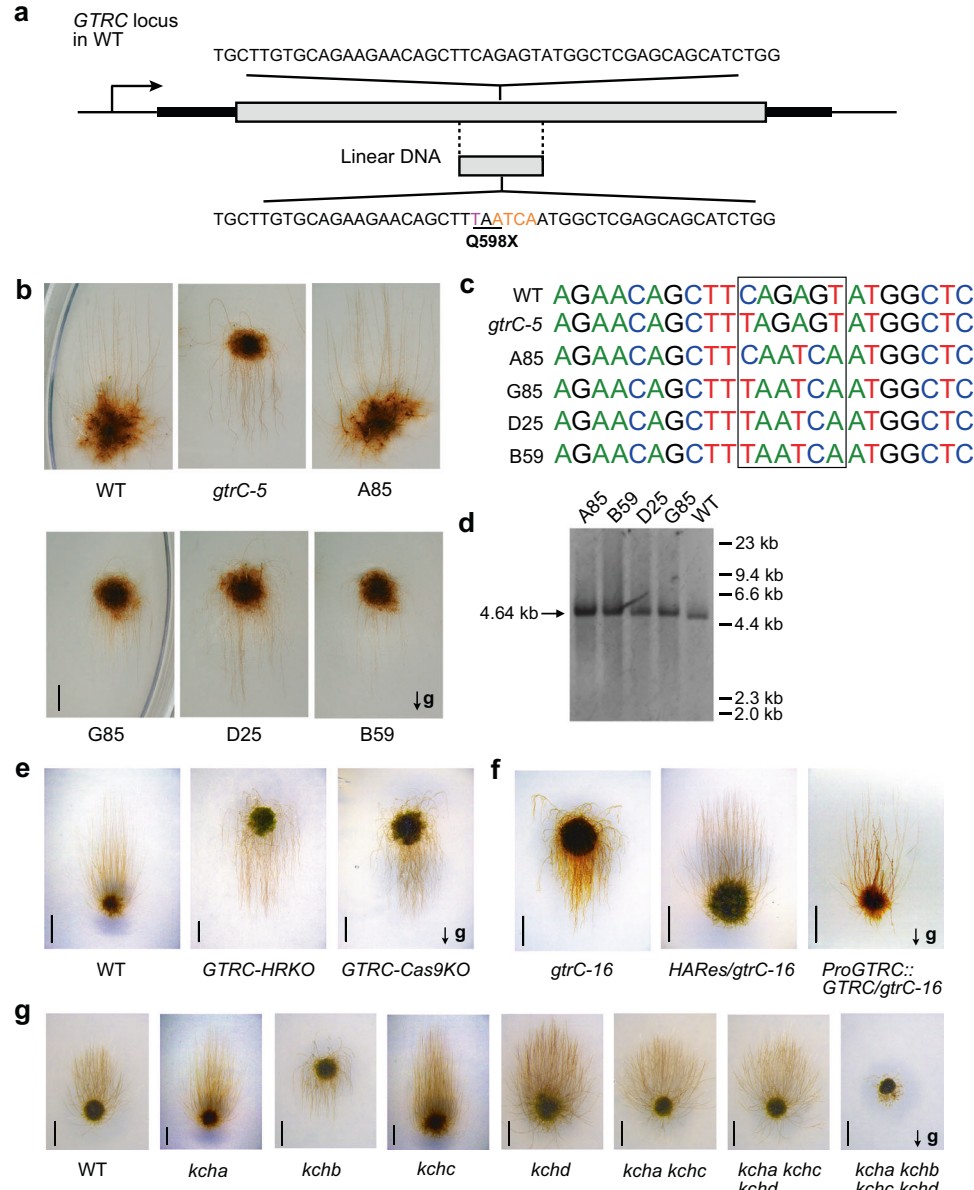

**Fig. 3 Genetic confirmation of the *GTRC* locus. a–d** Targeted mutagenesis of *GTRC* in wild type. **a** The native sequence encoding Q[598] was mutated to reproduce the stop codon (X) mutation identified in the *gtrC-5* mutant. WT *P. patens* was transformed with a linear 1537-bp fragment of DNA comprising the four exons surrounding the Q[598] codon in exon 14 using a transient selection strategy. The WT and mutant sequences are shown above and below. **b** Gravitropism in WT, *gtrC-5* and gene-targeted WT *P. patens*. Caulonemal growth in darkness for 3 weeks on vertical BCD agar medium. The untransformed WT and *gtrC-5* mutant lines are shown in the upper panel together with a transgenic line A85 that did not alter the Q[598] residue. The lower panel shows three transgenic lines (B59, D25, and G85) with a Q598X mutation. **c** Sequence analysis of the mutant locus in the gene-targeted lines, compared with WT and the original *gtrC-5* mutant. Lines B59, D25, and G85 contain the targeted mutation (CAGAGT > TAATCA), resulting in Q598X (CAG > TAA). Line A85 contains the targeted mutation of Ser[599] (AGT > TCA) but unmutated Q[598] (CAG > CAA). The nucleotides coding these two amino acids were labeled in the black box. **d** Southern blot analysis of the gene-targeted mutant lines. The genomic DNA was digested by *Eco* RI and then detected by labeled ssDNA. Only one band of the expected size (ca. 4.64 kb) was detected in each line, confirming that no off-target integration occurred. The experiment was carried out once, and the result is unambiguous. See methods for details, and source data are provided as a Source Data file. **e** Gravitropic phenotypes of knockout lines of *GTRC*. *GTRC-HRKO* was generated via homologous recombination by substituting *GTRC* genomic region with the NptII cassette. *GTRC-Cas9KO* was generated by editing the *GTRC* locus via CRISPR-Cas9 mutagenesis. **f** Gravitropic phenotypes of complementation lines of *GTRC*. *HARes/gtrC-16* was generated by substituting the mutated nucleotide of *gtrC-16* with wild-type genomic nucleotide. *ProGTRC::GTRC/gtrC-16* was generated by substituting the mutated genomic DNA of *GTRC* in *gtrC-16* with wild-type *GTRC* coding sequence. The details of the constructs and genotypes for the *P. patens* lines in **e**–**f** are shown in Supplementary Fig. 1. **g** Gravitropic phenotypes of knockout lines of *KCH* family genes. All lines except *kchb* (*gtrC-16*) were generated by editing the corresponding *KCH* loci via CRISPR/Cas9 mutagenesis. Genotypes for these lines are shown in Supplementary Fig. 3. In (**b**, **e**, **f**, **g**), arrows labeled with "g" indicate the directions of gravity vectors, and the scale bars are 3 mm.

Supplementary Fig. 1b). These complementation experiments unequivocally substantiate the gene's identity. Of the additional *gtrC* alleles obtained, *gtrC-17* is defective in splicing (Supplementary Fig. 1c), and *gtrC-18* harbors an amino acid change in the motor domain that could be complemented by *GTRC* (Supplementary Fig. 1d).

GTRC/KCHb is a member of the kinesin 14-II/KCH subfamily[29], and the kinesins in this subfamily play redundant roles in regulating cell growth[30]. Two phylogenetic trees based on motor domain or full-length protein sequences showed slightly different topologies (Supplementary Fig. 2). The tree based on the motor domain was identical to that obtained in previous studies[29,30], and both trees suggested that KCHd is distinct from the other three KCHs. We tested whether other members in this subfamily were also involved in gravitropism by generating CRISPR-mediated knockout mutants of the *KCH* subfamily (Supplementary Fig. 3). All the mutants (single, double, or triple) retaining *KCHb* showed no obvious gravitropic phenotype (Fig. 3g). When all four *KCH* genes were knocked out, the quadruple mutant *kcha kchb kchc kchd* showed the reversed gravitropic phenotype, as well as additional serious growth defects (Fig. 3g). Since a previous report showed that *KCH* genes also mediated nuclear positioning[30], we investigated whether nuclear positioning is involved in gravity signaling. Our analysis showed that nuclear localization in the *gtrC-16* mutant was the same as in wild type (Supplementary Fig. 4), which indicated that the role of GTRC/KCHb in gravitropism was not related to nuclear positioning. Together, these data demonstrate that GTRC, although showing redundant roles with other KCH factors in mediating nuclear positioning and growth, has a unique function in directing the negative gravitropism of *P. patens* protonemata, and loss of function of *GTRC* results in positive gravitropic growth.

Our study, more than 30 years after the initial description of gravitropic mutant alleles, defines *GTRC* identity. Further phenotypic characterization indicated that the phototropic responses of these mutants were the same as wild type, indicating that mutation of *GTRC* specifically disrupts gravity signaling, upstream of asymmetric growth pathways shared by phototropism and gravitropism (Supplementary Fig. 5).

**GTRC localizes to and moves on the microtubules**. To study the cellular localization of GTRC, the GTRC genomic sequence (from ATG) in *gtrC-16* was replaced by GFP N-terminally fused with the full-length GTRC coding sequence by homologous recombination (Supplementary Fig. 6a). To generate a *GFP-GTRC/RFP-TUB P. patens* line, the plasmid *mRFP-α-tubulin* (RFP-TUB)[31] was further transformed into the GFP-GTRC line. The protonemata of the *GFP-GTRC/RFP-TUB* transgenic line showed normal negative gravitropism (wild-type) in darkness, indicating that the fusion protein GFP-GTRC is functional. GFP-GTRC clearly localized on the microtubules (Fig. 4a), consistent with its property as a kinesin. Furthermore, oryzalin and latrunculin B treatments clearly showed that its localization specifically depended on the microtubules but not actin filaments (Fig. 4b and Supplementary Fig. 7). Then, we used spinning disk confocal microscopy to record the movement of the GFP-GTRC protein, and found that it always moved away from the tip along the microtubule (Fig. 4c and Supplementary Movies 1, 2). We calculated the migration speed of GTRC/KCHb to be 463 ± 122 nm/s (Fig. 4d), which was very close to the reported speed of KCHa migration (441 ± 226 nm/s)[30]. Since the plus ends of microtubules are oriented towards the apical cell tips in the polarized expansion zone[32], our data show that GTRC is a processive minus-end-directed kinesin, a eukaryotic motor protein[30], which defines the direction of gravitropic responses.

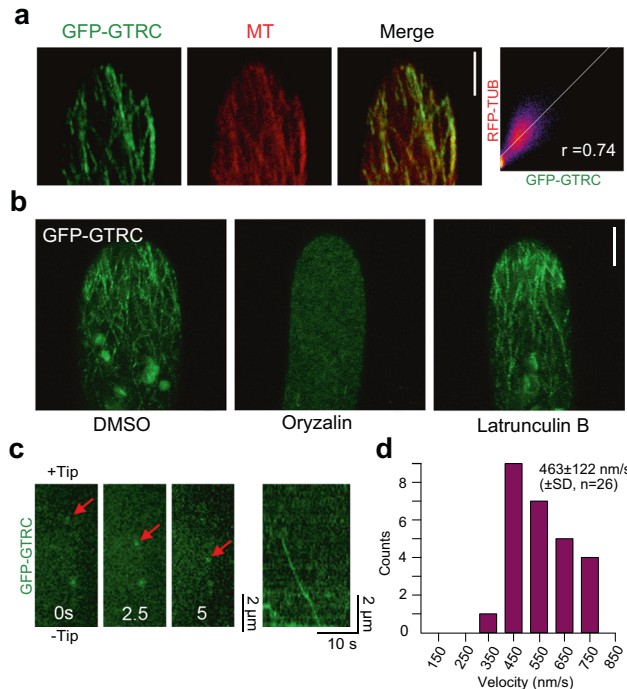

**Fig. 4 The localization and movement of GTRC proteins. a** Localization of GTRC on the microtubules (MT) in the tip cells of *P. patens* protonemata. RFP-TUB was co-expressed with GFP-GTRC in the protonemata, and fluorescence signals were collected by confocal microscopy. The scatterplots show the Pearson's correlation coefficient (*r*) between these two fluorescent signals. Scale bar, 5 μm. **b** Localization of GTRC was dependent on tubulin but not actin. Images were taken after 10 min of oryzalin or latrunculin B treatments, and oryzalin but not latrunculin B treatment disrupted the regular signal patterns of GTRC. Scale bar, 5 μm. More than 10 cells were studied for each treatment, and the results were similar. Oryzalin and latrunculin B disrupt the microtubules and actin filaments, respectively, as shown in Supplementary Fig. 7. **c** GTRC moves along MT in the minus-end direction. Left, processive movement of GTRC recorded by time-lapse photography using spinning disk confocal microscopy at 250 ms intervals. Red arrows mark the GFP-GTRC proteins moving along MT. Right, kymographs for the movement of GFP-GTRC. **d** Velocity of moving GFP-GTRC signals. The mean value is shown with standard deviation and examined sample size. Source data are provided as a Source Data file.

**Functional analyses of the domains of GTRC protein**. The point mutation of *gtrC-18* within the motor domain disrupts the function of GTRC, indicating the essential role of the motor domain (Figs. 1b, 2d). Further, GFP fused to various truncated *GTRC* fragments were used to replace the mutated genomic *GTRC* in *gtrC-16* by homologous recombination. The expression of these variants was confirmed by Western blot analysis (Supplementary Fig. 8), and the phenotypes and localization of GTRC proteins were observed (Fig. 5a, b and Supplementary Fig. 6a, b). These domain mutational analyses revealed that the N terminus (482 a.a.), the motor domain and the C terminus (373 a.a.) were all essential for the function of GTRC in gravity signaling, but that the CH domain is not (Fig. 5b). Thus, our data suggest that the potential actin-binding ability of GTRC is unrelated to its function in gravity signaling, although OsKCH2 (a homolog of GTRC in rice) was shown to transport actin filaments on the microtubule in vitro[33]. Interestingly, although both are essential for the function of GTRC in gravitropism, the C terminus (373 a.a.) is essential for the microtubule localization, while the N terminus (482 a.a.) is not (Fig. 5b).

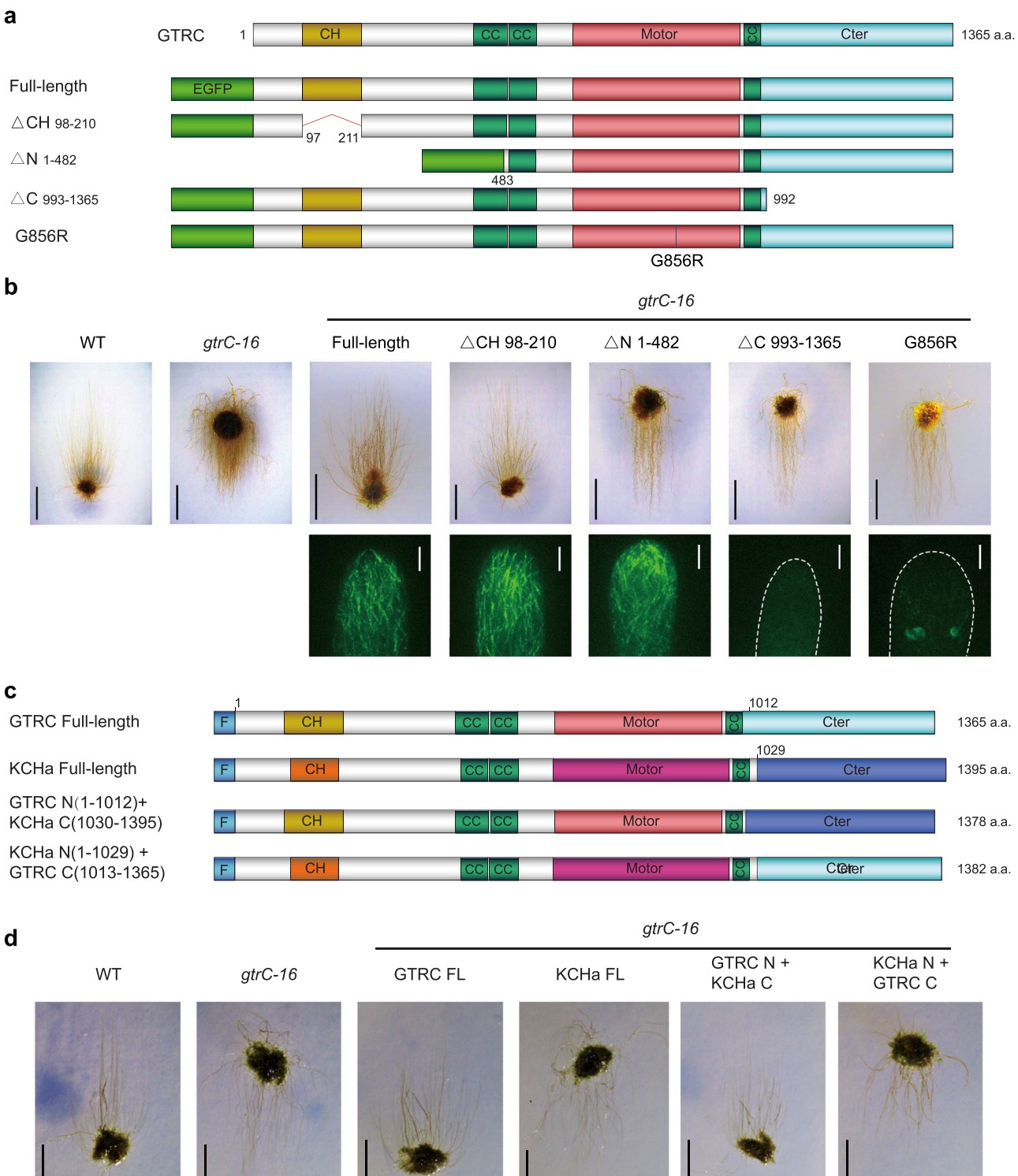

**Fig. 5 Functional domain analysis of GTRC protein. a** Schematic diagrams of full length and truncated constructs of *GTRC*. GFP was fused to the N-terminus of each fragment. **b** Complementation and fluorescence analyses of truncated GTRC. Upper, various constructs of *GTRC* were used to replace the genomic *GTRC* in *gtrC-16* via homologous recombination, and then the protonemata were grown on vertical plates. Scale bars, 3 mm. Lower, subcellular localization of GFP-fused full length or truncated GTRC proteins. Scale bars, 5 μm. White dashed lines depict the contours of the tip cells. For each genotype, the fluorescence of more than 10 tip cells was collected, and the results were similar. **c** Schematic diagrams of full length and chimeric constructs of *GTRC* and *KCHa*. 3×FLAG (F) was fused to the N-terminus of each fragment. **d** Functional analyses of chimeric proteins between GTRC and KCHa. Various constructs were used to replace the genomic *GTRC* in *gtrC-16* via homologous recombination, and then the protonemata were grown on vertical plates. Scale bars, 2 mm. The details of the constructs and genotypes for the *P. patens* lines are shown in Supplementary Fig. 6.

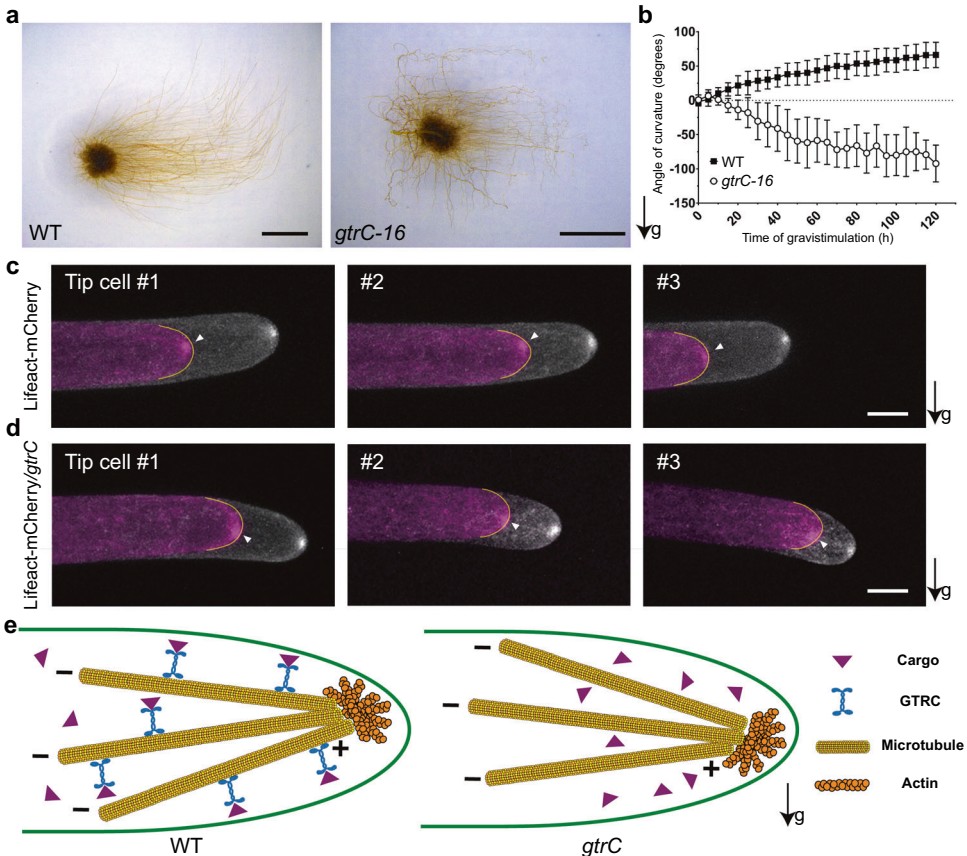

**Fig. 6 Disruption of GTRC reversed the gravity-triggered asymmetric distribution of actin filaments. a** Three-week dark-grown WT protonemata showed negative gravitropism and *gtrC-16* protonemata showed positive gravitropism after gravistimulation for 2 weeks on vertically orientated culture plate. **b** Kinetic curved angles of WT and *gtrC-16* filaments under gravistimulation (means ± s.d., n ≥ 18). Source data are provided as a Source Data file. **c** The apical cells of Lifeact-mCherry protonemata growing upwards under gravistimulation. Actin tended to accumulate at the upper part of tip region (white arrowheads). **d** The apical cells of Lifeact-mCherry/*gtrC* protonemata growing downwards under gravistimulation. Actin tended to accumulate at the lower part of tip region (white arrowheads). In (**c**, **d**), protonemata expressing Lifeact-mCherry were grown in the dark vertically for 1 week and then turned 90 degrees to be gravistimulated. Three representative tip cells were shown for each genotype, and the mCherry signals at two time points were merged to show cell elongation. Magenta and white represent earlier and later time points respectively, with an interval of around half an hour. Yellow lines depict the contour of tip cells at earlier time points. For each genotype, the fluorescence of 10 cells was collected. Arrows labeled with "g" indicate the direction of gravity vectors. Scale bars, 10 μm. **e** The model of the roles of GTRC in gravitropism. In WT protonemata, after gravistimulation, motor GTRC transfers unknown cargoes to the minus end of microtubules, resulting in the accumulation of actin filaments at the upper region of apical cell, which triggers negative gravitropism. In *gtrC* protonemata, deficiency of *GTRC* leads to abnormal cargo distribution, resulting in accumulation of actin filaments at the lower region of apical cell, which triggers positive gravitropism.

To dissect the domains determining the unique role of GTRC in gravitropism, full length or chimeric constructs between *KCHa* and *GTRC/KCHb* were fused with FLAG tags and used to replace the mutated *GTRC* in *gtrC-16* (Fig. 5c and Supplementary Fig. 6c, d). Western blot analysis showed that the expression levels of the KCHa, KCHb, and two chimeric KCH proteins were comparable (Supplementary Fig. 9). First, when both driven by *GTRC* promoters, the full-length *GTRC* coding sequence correctly complemented the mutant phenotype, but *KCHa* did not (Fig. 5d), which further demonstrated that their biological properties are distinct. Then, using the same strategy, we found that the chimeric protein with the GTRC N-terminus (1012 a.a) fused to the KCHa C-terminus (366 a.a.) could fully complement the *gtrC-16* mutant phenotype, while the reciprocal chimeric fusion (KCHa N-terminus (1029 a.a) to GTRC C-terminus (353 a.a.)) could not (Fig. 5d). Thus, the unique role of GTRC/KCHb among the *KCH* subfamily members in gravity signal transduction resides within its N-terminal (1012 a.a.) region, rather than the C-terminal (353 a.a.) domain. The specific amino acids and the underlying mechanism required for this unique role will be characterized in future studies.

**GTRC mediates the polarity of actin distribution to trigger bending under gravistimulation**. We also investigated how GTRC regulates gravitropism. Since a previous study showed that the actin cluster near the cell apex dictates the growth direction of the tip cell in moss protonemata[15], we examined whether the kinesin GTRC regulated the location of the actin cluster under gravistimulation. First, we recorded the continuous gravitropic response processes of both wild type and *gtrC-16*, the latter showing opposite bending compared to wild type (Fig. 6a, b). Since both wild type and *gtrC-16* started to show obvious bending during the first several hours, we checked the distribution of actin filaments under gravistimulation within this period. Excitingly, we found that the actin clusters concentrated at the upper halves of the wild-type cell apex when the filaments bent upwards, prefiguring the directional bending of the protonemata (Fig. 6c and Supplementary Fig. 10). In another earlier study, no major

redistribution of actin filaments was detected after gravistimulation, probably due to technical limitations at that time[22]. We further checked the distribution of actin in the *GTRC-HRKO* mutant with the expression of Lifeact-mCherry (named as Lifeact-mCherry/*gtrC*), and found that the actin clusters preferred to concentrate at the lower halves of the cell apex when the filaments bent downwards under gravistimulation (Fig. 6d and Supplementary Fig. 10). Thus, GTRC is required to direct the normal redistribution of actin filaments triggered by gravity.

## Discussion

The cytoskeleton has long been considered to be involved in gravitropism and most previous studies have focused on the actin filaments, the role of actin being believed to affect the sedimentation of amyloplasts in angiosperms[7–10]. In this study, we identified a microtubule-dependent motor (kinesin) factor, GTRC/KCHb, which is essential for establishing the correct polarity of moss gravitropism. This used a forward genetic strategy, in which we analyzed an historic mutant (*gtrC-5*) and a number of additionally generated and characterized new allelic *gtrC* lines. For this, two approaches were used to identify the *GTRC* locus: a map-based analysis of a segregating population and a next-generation sequencing strategy. The map-based identification of the *gtrC-5* mutant required two rounds of SNP genotyping before sequencing a candidate gene (Fig. 2a). This strategy is simple, but labor-intensive. The high-throughput sequencing used to identify *gtrC-16* required two runs of next-generation sequencing, that of the mutant itself and a mixture of segregants growing downwards (Fig. 2b), to generate a high-resolution SNP ratio map and directly identify the mutant locus. This is rapid and efficient but requires bioinformatic expertise. A similar high-throughput sequencing strategy has been used to map the microtubule depolymerizing-end-tracking protein CLoG1, that regulates microtubule dynamics[34]. These and other studies demonstrate the potential of the forward genetic strategy to identify components of complex developmental processes, when combined with the well-established reverse-genetic methodologies in *P. patens*[27,35,36].

The KCH subfamily members mediate protonemal growth and nuclear localization redundantly[30], whereas GTRC/KCHb showed a unique role in gravity signaling that is unrelated to nuclear localization (Fig. 3, Fig. 5 and Supplementary Fig. 4). This, together with the observation that the actin focus predicting the direction of growth of moss protonemata is re-oriented in the *gtrC* mutant, supports the involvement of both microtubules and actin filaments in establishing polar gravitropic growth, through minus-end directed movement of the GTRC kinesin. This provides a new insight into gravity signaling and highlights the power of a genetic approach to open the door for further mechanistic analysis of gravitropism. We propose a model to describe the possible roles of GTRC in gravitropism (Fig. 6e). In the protonemata of wild-type *P. patens*, GTRC motors transfer as-yet uncharacterized cargoes to the minus end of microtubules (away from the tip) after gravistimulation, which trigger the accumulation of the actin filament cluster at the upper region of tip cell apex. This accumulated actin likely further mediates the transport of secretory vesicles to the new growth point, driving the tip cell to grow upwards (negative gravitropism) (Left, Fig. 6e). In contrast, when *GTRC* is disrupted, the cargos are not transferred to the minus end of microtubules, and some unknown factors mediate the accumulation of actin filaments cluster at the lower region of the apical cell apex, which trigger the tip cell to grow downwards (positive gravitropism) (Right, Fig. 6e).

This raises questions that remain to be answered: what is the cargo and how does it interact with the actin cluster? What is the mechanism by which the orientation of gravitropic bending is determined? Since feedback regulation exists between microtubules and actin filaments in polarized growth[15], one explanation might be that the distribution of microtubule foci shows a similar asymmetric pattern as actin clusters during gravistimulation. We generated TUB-GFP and TUB-GFP/*gtrC* (Knocking out *GTRC* in TUB-GFP) lines using a previously reported plasmid[37], and found that the general microtubule organizations in wild type and in the *gtrC* mutant were similar (Supplementary Fig. 11). Under gravistimulation, for the protonemata of TUB-GFP lines showing negatively gravitropic responses, the microtubule foci clearly accumulated on the upper region of cell apex (Supplementary Fig. 12). Due to the severe growth defect of TUB-GFP/*gtrC* lines (Supplementary Fig. 11a), we could not examine how GTRC affects the distribution of TUB-GFP under gravistimulation. For studying the effect of gravity on microtubule distribution and dynamics, we also generated another line: p7113-mRFP-tub. However, this line showed a strong signal in the basal parts of the dark-grown protonema but a very faint signal in the tip cells (Supplementary Fig. 13). This required the use of a high-power laser and a long exposure time to collect the signal (Fig. 4a), rendering long-term and dynamic observations of microtubule impossible. Thus, we can only speculate that the tubulin foci locate at the lower region of cell apex in the *gtrC* mutant background under gravistimulation (Right, Fig. 6e).

Because the actin-binding CH domain appears to be dispensable for correct gravitropic bending, we rule out the possibility that GTRC directly transports the actin filaments to regulate gravitropism (Fig. 5a, b). Since myosin XI-associated structures anticipate and organize the actin polymerization machinery[20], a further possibility is that GTRC, microtubules, myosin XI, and actin play sequential roles in directing polar tip growth in gravitropism. This remains to be resolved, and potential experimental approaches involve the identification and characterization of the cargo transported by GTRC, and the identification of further regulatory components in genetic screening. Our initial screening also identified another locus (currently under investigation), that when mutated, also shows a downward gravitropic phenotype similar to *gtrC*. This is the first indication that a group of factors combine to direct the gravitropic polarity of moss protonema. Concomitantly, the analysis of other downward-growing mutants, such as those previously identified in *Ceratodon*[38], may also be expected to identify new factors participating in this regulatory network. When GTRC was mutated, the protonemata grow downwards along the gravity vector but the gravitropic response was faster compared to wild type (Fig. 6a, b), suggesting the existence of additional factors that transduce the gravity signal in the opposite direction compared to GTRC, whose roles are masked in the wild type. Together, the kinetics of protonemal gravitropic responses depend on the competence of these two groups of factors. The identification and functional analysis of these factors will further reveal the molecular mechanisms of gravity signaling.

Beyond its role in gravitropism in *P. patens*, it would be interesting to determine whether *GTRC* homologs also mediate gravitropic or polar growth of tip-growing cells in other plant species. Phylogenetic analysis shows that *GTRC* homologs are well-conserved in multiple organisms (Supplementary Fig. 14), raising the possibility that these GTRC homologous proteins may play conserved roles in the evolution of gravitropic responses—important adaptive traits for the colonization of land and the establishment of an erect plant body. The protonemal gravitropic system, which has been a highly amenable experimental model in this study, is likely also to reveal general mechanisms about directional polar growth in other types of cells in which

microtubule motors have been implicated. These include pollen tubes[39], root hairs[40] and trichome architecture[41] in flowering plants, hyphal growth in filamentous fungi[42], and vertebrate neurons in which axonal growth and long-range anterograde and retrograde transport is mediated by kinesins and dyneins, respectively[43,44].

## Methods

**Plant materials and growth conditions**. The Gransden (Gd) wild-type strain of *Physcomitrella patens* was used for most of the experiments, and the genetically divergent wild-type Villersexel K3 (Vx) strain was used for crossing with gravitropic mutants in genetic mapping. For regular growth, *P. patens* was cultured on BCD or BCDAT medium with 1% (w/v) agar and 1% (w/v) sucrose at 25 °C under continuous white light. For mutant screening, the protoplasts were resuspended in PRMT medium (BCDAT medium supplemented with 8% (w/v) mannitol, 50 mM CaCl$_2$ and 0.6% (w/v) agar). For collecting the phenotype in the dark, *P. patens* was cultured on BCD or BCDAT medium with 1% (w/v) sucrose and 0.5–1% (w/v) agar at 25 °C under continuous white light for around a week, and then the culture plates were kept vertically and moved into darkness for the indicated times at 25 °C. gtrC5 was described previously[24], and is now denoted *gtrC-5* in accordance with the nomenclature applied to new mutant alleles.

**Genetic mapping**. To map the mutations corresponding to the phenotypes, linkage analysis was performed by SNP analysis (Leeds) and deep sequencing (Beijing). In Beijing a single mutant was identified first, and then other mutants were checked for mutations within the same gene. After identifying a nonsense mutation in the first gene, and finding that the *gtrC-5* mutant also had a nonsense mutation within the same gene, the new mutant was named *gtrC-16* (Figs. 1b, 2c), since a series of *gtrC* mutants had been obtained in the previous studies although the gene had not been identified[25]. For other *gtrC* mutants, a mutant affecting the splicing 5' to the nonsense mutation in *gtrC-16* was named *gtrC-17* (Figs. 1b, 2c, and Supplementary Fig. 1c). Another mutant *gtrC-18* with a nonsynonymous mutation in the same gene was confirmed by complementation assay (Figs. 1b, 2c, and Supplementary Fig. 1d). The genetic mapping processes for *gtrC* mutants were as follows.

First, the candidate genes were mapped to a region via linkage analysis. To generate segregating populations, *gtrC-16* (Beijing) and *gtrC-5* (Leeds) were crossed with wild-type Villersexel K3 (Vx). Plants were grown together on BCD medium (without ammonium tartrate, and concentration of KNO$_3$ was reduced to around 0.5 mM) at 25 °C under continuous white light for a month, and then the materials were moved to 16 °C under short-day conditions (8 h white light and 16 h dark) and irrigated until sporangia matured. The Gransden (Gd) wild type that we used for the mutant screening exhibited reduced male fertility, so that the first sporophytes to form on the Gd plants were hybrids, which was confirmed by genotyping segregants. The original Leeds *gtrC-5* mutant also carried an auxotrophic PABA-requiring mutation, so that the hybrid nature of the sporophytes was rapidly revealed by a 1:1 segregation of progeny on PABA-deficient medium. Mature spore capsules were washed and gently crushed in 1 ml sterile water in a tube to re-suspend the spores, and then the spores were kept a week at 4 °C for high germination rates. After the germination of spores, the colonies were picked out and grown in the dark for segregation analysis (Fig. 2a, b). Segregants were selected as individuals for SNP mapping (Leeds), or those growing downward were combined and the DNA was extracted for whole-genome sequencing (Beijing).

SNP mapping of the *gtrC-5* mutation comprised two steps. A small number of segregants was included in the populations subjected to GoldenGate bead array SNP mapping for construction of the *P. patens* chromosome-scale linkage map[45]. This defined an interval of *ca.* 840 kb on chromosome 2 (Fig. 2a). This was refined by allele-specific PCR-genotyping of 6 SNP loci and a single SSR locus within this interval in a population of 282 segregants to identify a region of 238 kb. Sequencing of the candidate gene *Pp3c2_9150* identified a C > T transition causing a premature termination codon (CAG > TAG) (Fig. 2c).

DNA-Seq and read mapping for analyzing *gtrC-16* were performed as described below. Sequence data of Villersexel were retrieved from NCBI SRA (SRX030894). Single nucleotide polymorphisms (SNPs) used as genetic markers in linkage analysis were called by GATK in gvcf-mode on specific genomic intervals defined by SNPs between Gd and Vx. Allele depth was extracted from SNPs of segregants, and used to calculate the Gd/Vx ratio. The ratio was plotted in R to detect peaks where the ratio approached 1.0.

Then, the mutations along the whole genome were analyzed in *gtrC-16*, and those located in the peak regions identified above were selected. Total DNA was extracted from *gtrC-16* using the CTAB method with RNase digestion. Genome resequencing was performed on a NovaSeq 6000 Sequencing System (Illumina, Inc., San Diego, CA, USA), generating more than 70 million 150-bp paired-end reads per sample, calculated to generate 50x coverage of the *P. patens* genome. Reads were viewed in fastqc (https://www.bioinformatics.babraham.ac.uk/projects/fastqc) to assess quality and trimmed by seqtk (https://github.com/lh3/seqtk).

Trimmed reads were mapped to the *P. patens* V3.3 genome (DoE-JGI, http://phytozome.jgi.doe.gov) using bwa-mem[46]. Variant calling was performed with SAMTOOLS[47] and GATK[48]. The variant calling by SAMTOOLS used MPILEUP and BCFTOOLS with default parameters. The GATK pipeline with Best Practices was used, with the modification that the quality recalibration step was omitted. After filtering low-quality variants, a.vcf file was generated by CombineVariants in GATK and annotated with ANNOVAR[49]. Only nonsynonymous and nonsense mutations within genes were retained. Further analysis identified one nonsense mutation in *Pp3c2_9150* within the mapped peak region of *gtrC-16* (Fig. 2b).

**Generation of CRISPR system for *P. patens***. To integrate Cas9 into the *P. patens* intergenic (PIG) region, two adjacent fragments PIG1b L and PIG1b R described previously[50], were amplified from *P. patens* genomic DNA using primers PIG1b L_XbaI F/PIG1b L_SacI R and PIG1b R_KpnI F/PIG1b R_HindIII R, respectively. PIG1b L and PIG1b R were inserted into the *Xba* I/*Sac* I and *Kpn* I/ *Hin*d III restriction sites of pHIZ2, respectively, yielding the vector pHIZ2-PIG1b R-PIG1b L. A Cas9 expression module was amplified from p2×35S-Cas9-At[51] using primers 35 S_HindIII F and Tnos_NotI R, and inserted into pHIZ2-PIG1b R-PIG1b L to generate pHIZ2-PIG1b R-Cas9 cassette-PIG1b L (abbr: pHIZ2-Cas9), which was further digested by *Kpn* I and used to transform wild type *P. patens* (Gd) via homologous recombination to generate stable Cas9 expression lines.

To generate a single-guide RNA expression module, restriction site *Sap* I of a pBS plasmid was mutated with primers pBS-Modified F and pBS-Modified R. PpU6 was amplified from *P. patens* genomic DNA with primers PpU6 F and PpU6 R, and inserted into *Spe* I and *Bbs* I digested pAtU6-SK using a Gibson assembly kit[51]. PpU6-Scaffold was inserted into the *Kpn* I and *Sac* I restriction sites of modified pBS plasmid using primers PpU6_KpnI F and Scaffold_SacI R, generating plasmid pBS-PpU6-Scaffold. The vector pBS-PpU6-Scaffold was digested with *Bbs* I and ligated with the annealed fragment amplified by primers SapI F and SapI R, to construct plasmid pBS-PpU6-SapI-NotI-XhoI-SapI-Scaffold. The *ccdB* expression module was inserted into the *Not* I and *Xho* I restriction sites of pBS-PpU6-SapI-NotI-XhoI-SapI-Scaffold using primers ccdB_NotI F and ccdB_XhoI R, producing pBS-PpU6-ccdB-Scaffold. The expression cassette *35S:HPT-Tnos* was cloned from pTN186 with primers HPT_HindIII F and HPT_SacI R and inserted into *Hin*d III and *Sac* I digested pBS-PpU6-ccdB-Scaffold to generate plasmid pBS-PpU6-ccdB-Scaffold-35S:HPT-Tnos (abbr: pBS-sgRNA). The forward and reverse primers of each sgRNA, designated as sgRNA-F and sgRNA-R, were annealed and cloned into pBS-sgRNA digested with *Sap* I to make sgRNA constructs. For gene knockout experiments, sgRNA constructs were transformed into Cas9 stable expression lines, or co-transformed with pHIZ2-Cas9 into wild type or other genotypes.

**Generation of mutants and transgenic lines in *P. patens***. To construct the *GTRC-HRKO* line, *P. patens* genomic DNA was used as the template. The right homologous arm was amplified via PCR using the primers FE2KO-RF1 and FE2KO-RR1, and was then digested and inserted into the *Xba* I and *Sac* I sites of the pTN182 plasmid. The left homologous arm was amplified using the primers FE2KO-LF1 and FE2KO-LR1, and was then digested and inserted into the *Kpn* I and *Cla* I sites of the former plasmid to make the GTRC-HRKO plasmid. GTRC-HRKO plasmid was then transformed into wild-type *P. patens* or a Lifeact-mCherry[20], to generate *GTRC-HRKO* line and *Lifeact-mCherry/gtrC* line, respectively.

To construct a *gtrC-5* mutant in a WT Gd background, *P. patens* DNA was amplified with the primers KHC_MutS and KHC_A7. The amplicon was used as a megaprimer in an overlap amplification of genomic DNA, in combination with primer KHC_S6 and the amplicon was cloned in the *Eco*R V site of pBS SK$^+$ and sequenced to verify mutagenesis generating a Q$^{598}$Ter mutation. This was used as the template to amplify a 1537-bp fragment with primers KHC_KIS and KHC_KIA, and the amplicon was used to transform protoplasts in combination with the circular selection plasmid pMBL5 for transient selection. Transformants were selected on a medium containing G418 and explants transferred to drug-free medium for 2 weeks to ensure the loss of the selection vector. In successful homologous recombinant lines, linear DNA was expected to substitute wild-type genomic sequence, generating a Q$^{598}$Ter mutation (Fig. 3a). Allele-specific PCR was undertaken to distinguish between gene-targeted mutants (primers KHC_S6 and KHC_mut_A) and untargeted regenerants (primers KHC_S6 and KHC_WT_A). Single-copy allele replacements were detected by amplification with the external primers KHC_S6 and KHC_A7 and verified by sequence analysis. Since transformation in *P. patens* sometimes causes multi-site insertion, a Southern blot assay was performed to confirm no off-target integration occurred. Genomic DNA was digested by *Eco* RI and run on a gel (0.7% agarose). If no off-target integration occurred, we expect to detect a single DNA fragment of 4.64 kb, released from *Eco*R I sites at positions 6,376,867 and 6,381,509 in chromosome 2. Blot transfer to Hybond N + membrane (Merck GERPN203B) followed standard procedures (depurination: 0.25 M HCl, 15 min; denaturation: 1.5 M NaCl, 0.5 M NaOH, 30 min; neutralization: 1.5 M Tris-HCl, 1.5 M NaCl, 1 mM Na$_2$EDTA, pH 7.4, 30 min; transfer: 20x SSC, overnight). DNA was UV-crosslinked to the membrane using a Stratalinker. For detection, the probe sequence was labeled by PCR amplification of the 1537-bp targeting fragment using primers KCH_KIS and

KCH_KIA in a reaction containing 200 μM dATP, dCTP, and dGTP, and dTTP at 140 μM and digoxygenin-labeled dUTP at 60 μM. Prehybridization and hybridization used DIG Easy Hyb buffer (Roche 11603558001), prior to successive washes in 2x SSC (0.1% SDS) and 0.5x SSC (0.1% SDS) at 65 °C. Hybridizing sequences were detected using the DIG luminescent detection kit (Roche 11363514910). If there were unexpected substitutions or insertions, more than one band would be detected. Lines confirmed as containing a single copy of the altered gene were verified by DNA sequence analysis and the gravitropic phenotype was analyzed in a growth test.

To construct *GTRC*, or *GFP-GTRC* (full length or truncated) lines driven by the *GTRC* native promoter, the flanking sequence downstream of the terminal codon of *GTRC* was amplified by csKCH_RF and csKCH_RR from the *P. patens* genome, digested with *Xba* I and *Sac* I, cloned into pTN182, generating the plasmid pCskch_R. The GTRC promoter was amplified by csKCH_LF1 and csKCH_LFIR. The terminal element was amplified by csKCH_LFIF and csKCH_LR1 from the pJim19 plasmid. Together, the left arm containing a promoter, multiple cloning sites, and a Nos terminator was amplified by overlap extension PCR (OEPCR) using primers csKCH_LF1 and csKCH_LR1, and inserted into pCskch_R after being digested with *Kpn* I and *Cla* I, generating the plasmid pCskch. The GTRC coding sequence was amplified from *P. patens* total RNA by RT-PCR with GTRCcdsF and GTRCcdsR, digested with *Not* I and *Nhe* I, and cloned into the *Not* I and *Nhe* I sites of pCskch, generating the plasmid pCskch-GTRC. The GFP-GTRC fusion coding sequence was amplified by OEPCR from the GFP coding sequence (Amplified by GFP-GTRCF and GFP-GTRCIR from PJim19-GFP) and the GTRC coding sequence (amplified by GFP-GTRCIF and GFP-GTRCR from pCskch-GTRC), digested with *Not* I and *Nhe* I, and cloned into pCskch, generating pCskch-GFP-GTRC. Various versions of truncated GTRC fused with GFP were generated from pCskch-GFP-GTRC by OEPCR through the same restriction enzyme site with different overlapping primers. The overlapping primers for GTRC-DeN were GD-deN_IR and GD-deN_IF. The overlapping primers for GTRC-DeCH were GD-deCH_IR and GD-deCH_IF. GTRC-DeC was amplified with GFP-GTRCF and GD-deC_R.

To generate chimeric constructs of *GTRC* and *KCHa* fused with FLAG tag, pCskch was used as the basic plasmid. Sequence encoding the N terminus (1-1012 a.a.) of GTRC was amplified by aCbN_F and aCbN_IR, and sequence encoding the C terminus (1030-1395 a.a.) of KCHa was amplified by aCbN_IF and aCbN_R. Then, these two fragments were used as the templates for OEPCR using primers aCbN_F and aCbN_R, and the resulting chimeric fragment was inserted into *Not* I and *Nhe* I digested pCskch, generating pCskch_bNaC. Sequence encoding the N terminus (1-1029 a.a.) of KCHa was amplified by aNbC_F and aNbC_IR, and sequence encoding the C terminus (1013-1365 a.a.) of GTRC was amplified by aNbC_IF aNbC_R. Then, these two fragments were used as the templates for OEPCR using primers aNbC_F and aNbC_R, and the resulting chimeric fragment was inserted into *Not* I and *Nhe* I digested pCskch, generating pCskch_aNbC. For controls, *GTRC* was amplified by GTRC_F and GTRC_R, and *KCHa* was amplified by KCHaF and KCHaR, and the fragments were cloned in pCskch to generate pCskch_GTRC and pCskch_KCHa respectively. The left part of 3×FLAG fragment (including sequences for recombination and linker) was amplified by LarmF and 3F-IR from the plasmid pCskch. The right part of 3×FLAG fragment was amplified by 3F-IF and 3F-notR from the plasmid pCskch-GFP-GTRC. Then, these two fragments were used as the templates for OEPCR using primers LarmF and 3F-notR, and the resulting FLAG fragments were inserted into *KPN* I and *Not* I digested pCskch_bNaC, pCskch_aNbC, pCskch_GTRC, and pCskch_KCHa, generating pCskch_FGbNaC, pCskch_FGaNbC, pCskch_FGGTRC, and pCskch_FGKCHa respectively.

To construct the tubulin marker line, pGFP-257/pTN90 was digested by *Not* I for transformation. Plasmid p7113-mRFP-tub was digested by *Kpn* I for transformation[31]. To generate the GFP-GTRC/RFP-TUB line, pCskch-GFP-GTRC and p7113-mRFP-tub were used to co-transform the *gtrC-16* mutant.

For generating CRISPR/Cas9 knockout lines, protoplasts were co-transformed with constructs containing Cas9 and sgRNAs. Stable transgenic lines were then grown to the gametophore phase, and several pieces of leaves were cut for generating material with clear genotypes. Individuals after first-round screening were genotyped by PCR.

Unless otherwise specified, the final constructs were digested with *Kpn* I to provide linear DNA for transformation via homologous recombination[52]. Around one-week-old white light-grown protonemata were digested with 1% driselase (w/v) (Sigma) in 8% (w/v) D-mannitol for 1 h. The protoplasts obtained were washed twice with 8% mannitol and resuspended at a concentration of 1 to 1.5×10^6/ml in MMM solution (9.1% (w/v) D-mannitol, 15 mM MgCl$_2$, 5 mM MES (pH 5.6)). 20 μg linearized plasmid DNA (up to 30 μl volume) were mixed gently with 300 μl protoplast suspension and 300 μl PEG solution (0.1 M Ca(NO$_3$)$_2$, 8% (w/v) D-mannitol, 40% (w/v) PEG 6000, 10 mM Tris (pH 8.0)). The mixture was incubated at room temperature for 30 min. Heat shock at 45 °C for 5 min was optional. Then, samples were diluted with 1.5 mL W5 solution (2 mM MES (pH 5.6), 5 mM KCl, 125 mM CaCl$_2$, 154 mM NaCl) and resuspended in 8 ml PRMT medium (0.4% Agar, 10 mM CaCl$_2$, 5 mM diammonium tartrate, 6% D-Mannitol). Suspension was dispensed on PRMB medium (0.7% Agar, 10 mM CaCl$_2$, 5 mM diammonium tartrate, 6% D-Mannitol). Subsequent resistance screening was performed after 3-4 days after protoplasts regenerated.

**Cytoskeleton inhibitor treatments**. For inhibitor treatments, oryzalin (Yuanye Bio-Technology) and Latrunculin B (Abcam) were dissolved in dimethylsulfoxide (DMSO, Sigma) as stock solutions (100 mM). Final concentrations were 25 μM for oryzalin, and 50 μM for Latrunculin B. The final concentration of DMSO was < 0.1% (v/v). Moss was cultured and observed in confocal dishes with a thin layer of BCDAT medium, and inhibitors were directly added to the dishes. Cells were observed at the indicated times.

**Microscopic observation**. For the preparation of the cells for observation, 2 ml BCDAT medium with 1% (w/v) agar and 1% (w/v) sucrose was added into a glass-bottom dish (Nest Scientific 801001). After the medium solidified, a central core was excised and replaced by a thin layer of the same medium just covering the glass base. A small number of protonemata was inoculated onto this thin layer and cultured at 25 °C under continuous white light for around 4 days. The dishes were then kept vertically and moved into darkness for the indicated times at the same temperature.

For phenotypic observations, images were taken using stereo microscopes (M205FA; Leica). The fluorescent images were collected using a confocal laser microscope (Zeiss LSM800) unless otherwise indicated. The excitation laser wavelengths were as follows: GFP, 488 nm; mCherry, 561 nm; DAPI, 405 nm. For nuclear observation, protonemal cells were stained with DAPI (1 μg ml$^{-1}$). To observe the movement of GTRC-GFP along the microtubules, images were taken using spinning disk confocal microscopes (ZEISS, CSU-X1; Andor Dragonfly).

For fluorescence observation of the protonemata under gravistimulation, the fluorescent images were collected using a confocal laser microscope (Zeiss LSM800) with a vertical objective table. A 561 nm laser was used to detect the Lifeact-mCherry fluorescence. The protonemata were cultured in glass-bottom dishes that were fastened vertically for gravistimulation. Images are maximum projections of z-stacks acquired every 3 min. Fluorescent images were processed by ImageJ.

For analyzing the colocalization of GFP-GTRC and RFP-TUB, confocal images of protonema tip cells were analyzed using the ImageJ Coloc 2 to produce scatterplots. The scatterplots were obtained by analyzing the pixels inside the cell, resulting in Pearson's correlation coefficient (r) based on the automatic threshold.

**Protein extraction and Western blot**. Protonemata of *Physcomitrella patens* were collected and homogenized by TissueLyser (Retsch). Proteins were extracted by RIPA buffer (R0278, Abcam) with protease inhibitor Cocktail (MF182, Mei5bio), heated at 100 °C for 10 min, centrifuged at 15000 rpm for 10 min, and analyzed by Western blot. Proteins were separated in 8% SDS-PAGE, and transferred to PVDF membranes (Millipore). Membranes were blocked with 5% non-fat milk in PBST for 1 h, blotted with the primary antibody in 3% BSA overnight, washed 10 min × 3 times with PBST, blotted with secondary antibody in 5% non-fat milk, washed 10 min × 3 times with PBST and then incubated with ECL solution (GE Health-care). For detecting GFP-fused protein, the primary antibody was Anti-GFP (1:1000, cat # ab6556, Abcam) and the secondary antibody was anti-Rabbit-AP (1:5000, cat # 656122, Invitrogen). For detecting FLAG-fused protein by enhanced chemiluminescent (ECL) method, the primary antibody was Anti-FLAG (1:1000, cat # F1804, Sigma) and the secondary antibody was anti-mouse-HRP (1:5000, cat # A9044, Sigma).

**Phylogenetic analysis**. For generating the phylogenetic tree for Kinesin 14-II homologs, the *P. patens* and *Arabidopsis* 14-II protein sequences were used as templates to perform BLASTP against peptides annotated in 16 selected plant genomes with an E-value threshold of 1e−3. Domain prediction was performed using InterProScan (v5.24), and peptide sequences assigned with "Kinesin motor domain" annotation were screened out as kinesin candidate orthologs[53]. Candidate sequences were aligned with MAFFT, and positions with more than 50% gaps were removed using the Phyutility program (v2.2.6)[54,55]. ProtTest (v3.4.2) was used to select the best substitution model[56]. The maximum likelihood (ML) method implemented in FastTree (v2.1) was used to construct the phylogenetic tree of kinesin candidates based on the LG model under a GAMMA rate distribution with 500 bootstraps[57]. The extremely long branches were removed manually. Sequences belong to the monophyletic group containing *P. patens* and *Arabidopsis* kinesin 14-II sequences were realigned and trimmed to reconstruct the phylogenetic tree.

## Data availability

Data that support the findings of this study are available within the paper, and its Supplementary information files, or from the corresponding author upon reasonable request. The accession numbers for the whole-genome DNA sequencing reads reported in this paper are NCBI: PRJNA680887 and BIG GSA: CRA003554. The data deposited in these two databases are the same. Source data are provided with this paper.

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

## Acknowledgements

The authors thank Prof. Xing Wang Deng for lab resource and helpful discussion. Prof. Mitsuyasu Hasebe and Prof. Yuji Hiwatashi for sGFP-tubulin, mRFP-tubulin, pHIZ2, pTN182 and pTN186 plasmids, Prof. Luis Vidali for Lifeact-mCherry line in *P.patens*, Prof. Jean-Pierre Zryd for the BS213 plasmid, Prof. Ichiro Mitsuhara for the 7113 promoter, and Prof. Lei Li, Prof. Xiaowei Chen and Prof. Dongyi Xu for confocal microscopy. This work was supported by the National Key R&D Program of China

(2018YFE0204700, 2017YFA0503800), the National Natural Science Foundation of China (32022005, 31621001), and the Tsinghua-Peking Center for Life Sciences. A.C.C. and Y.K. acknowledge the support of the UK Biotechnology and Biological Sciences Research Council (Grant No: BB/F001797/1).

## Author contributions

H.C. designed and supervised the study; Y.L. and Z.D. mapped the *GTRC* gene in *gtrC-16* and carried out most of the experiments; Y.K., D.J.C., and A.C.C mapped and characterized the *GTRC* gene in *gtrC-5*; Z.C. obtained *gtrC-17* and *gtrC-18* via mutagenesis screening, and generated the mutants knocking out *KCH* family members; J.W. generated the CRISPR system in *P. patens*; X.H. and H.H. constructed the phylogenetic trees; Y.W. recorded the movement of GTRC-GFP on microtubule; H.C., A.C.C., D.J.C., Y.L., Z.D., W.T. interpreted the results and wrote the manuscript.

## Competing interests

The authors declare no competing interests.
