## [Peer Review File · Nature Communications]

Reviewers' comments:

Reviewer #1 (Remarks to the Author):

the major claim by Li et al is the identification of a gene for required for defining the direction (up versus down) of gravity response. When mutated the loss of this gene function leads to a reversed direction of protonemal gravitropism in the *gtrC* mutant. The authors claim that this gene is a KCHb that was previously reported as a member of kinesin-14 regulating nuclear transport and cytoskeletal coalescence for tip growth. They indicate that phenotype associates with a defined lesion and confirm this by showing that the phenotype can be repaired by introducing a wild type copy of the gene by transformation. They further confirm gene identity by KO of the candidate by 2 independent means. This aspect of the work seems very secure. Using a variety of other defined mutations and inferring from what is known about the kinesin gene family, the authors argue that the motor domain is necessary for function and show that the GFP fusion protein localises in a MT dependent manner. All these findings are consistent with its being a functional kinesin.

The manuscript is well organised and the finding is interesting in the research field on gravity sensing/response since this work reveals a mechanism whereby these cells can translate a directional signal into the growth response of the correct direction. Therefore, it should be of interest to quite a wide readership.

There are some concerns about some of the detailed mechanistic conclusions, which should be addressed:

1. Based on the apparent lack of MT localisation of the truncated KCHb fused to GFP, the C terminus is proposed to be essential for MT localisation. Indeed, it seems that no KCHb dependent GFP signals are detected in fig.5b. However, is the truncation of the C terminus likely to lead to protein instability and thereby rapid degradation, they should check the level of the GFP-fused protein (e.g. WB with anti-GFP antibody).
2. It is claimed that KCH b has a unique function in gravity signaling because no roles of both KCH a and c were not detected. How about KCHd? What are the gravi-phenotypes of double a/c or other combinations. These should be easily checked as all four of members of the family have been KOed?
3. KCHa did not complement the mutant phenotype so the authors claim that the biochemical activity of KCHb is different from those of KCHa. This claim is weak since no direct biochemical data is presented in the manuscript. The complementation test basically provides the biological property of the kinesin gene, not directly its biochemistry and there are many other reasons for lack of complementation.
4. It is proposed that the predominant function of the KCHb's N-terminal region is in gravity sensing. But the C-terminal region, as well as the N-terminal region, is essential for sensing and responding to gravity (fig.5a, b). The reciprocal domain swapping test between KCHb and KCHa suggest that the N-terminal region of KCHb is more divergent from that of KCHa, but again protein levels need to be checked to eliminate the more obvious confounding factors (such as protein instability).
5. The importance of an apically located actin cluster is suggested but the supporting evidence is quite weak and lacks sufficient quantification. The actin cluster accumulates at the upper region of the apical dome of wild type in response to the direction of gravity in wild type, while the MT foci did not. It is a bit curious because the AF cluster co-localises to the MT foci in the expanding apical dome under normal growth condition. This implies that, during the gravitropic growth response, the cytoskeletal organisation, or dynamics, within the apical dome is different from normal growth? If so, the effect of gravity on MT distribution and the MT dynamics both need quantification - are they altered in the apical dome and does the mutation affect either? In the Fig.6e schematics, the MT foci preferentially accumulate at the upper part of the apical dome, together with the AF cluster. This should be easy to clarify.

6 the lack of role in nuclear positioning is weakly supported, with only a pair of images (Supplementary figure 6 - the mutant nucleus seems displaced towards the tip but its significance is impossible to assess). L222 and 223 refers to evidence for this in Fig5 and supp fig 6 but i dont see any in Fig 5? Quantitative data is required to support this nuclear position claim, and ideally also from double and triple mutants if available.

Other minor points include:

1. P6, L135-136. The minus-end-directed movement should be assessed with MT dynamics that is associated with KCHb .
2. P10, L255. It should be added in the "Plant materials and growth conditions" section how gravitropic assay is carried out (e.g. medium and cultivation condition). BCD medium for this assay is correct? In dark condition, the medium is sometime supplemented with glucose or sucrose as a carbon source.
3. P10, L261. "50mM" may be replaced with "50 mM".
4. P13, L347. "Xba I" may be replaced with "Xba I".
5. P13, L347. "PTN182" may be replaced with "pTN182".
6. P13, L356. "1537bp" may be replaced with "1537-bp".
7. P14, L365. "1537bp" may be replaced with "1537-bp".
8. P15, L413. "100mM" may be replaced with "100 mM".
9. P15, L417. The preparation of the cells for observation is described in the "Microscopic observation" section. E.g. medium? What type of glass-bottom dish type was used?
10. P25, fig.3d. unfortunately, the southern blot is confusing. Please provide more information on this blot.
11. P35, L704. "Supplementary Fig. 1d" is incorrect? Please mention the fig that shows the primer position.
12. P37, L720 and 723 (supplemental fig.4 b and d). These panels look agarose-gel electrophoresis..., perhaps it's not DNA-gel blot analysis.

Reviewer #2 (Remarks to the Author):

The manuscript by Li et al. presents a thorough characterization of a reversed response gravitropic mutant of *Physcomitrella patens*. Although mutants that have this reversed response were initially isolated more than 30 years ago, the identity of the genetic lesion was unknown until now. The fact that the mutated gene is a kinesin that belongs to a subfamily with four members is surprising and reveals an exciting type of subspecialization of kinesin function. The observation that the apical actin cluster is positioned at the opposite side in the mutants is puzzling, and understanding the mechanism of this will prove to be very important for our future understanding of signaling directed to the tip-growth machinery.

Major comments:

- The title, as it is, is confusing and could be more appropriate for a review article. I suggest crafting a title so that it clearly states the content and conclusions of the manuscript.
- The descriptions for the mapping of the mutation are somewhat confusing. For example, it is not clear how the spores resulting from the outcross were selected from the ones resulting from selfing. It will also be useful to compare the two mapping approaches used for efficiency, cost, complexity, and time required.
- The lack of a nuclear positioning phenotype is not surprising, but the analysis, as presented, is not thorough. Only one cell is shown in supplementary Fig. 6, and no quantification of nuclear positions that could rule out a partial phenotype is present for the mutant.
- Similarly to above, the lack of MT organization phenotype at the tip is not thoroughly characterized. Only one image is presented as supplemental figure. This is a good quality image, but because microtubules have been shown to help position the apical actin cluster, it will be helpful to characterized the microtubule organization at the tip in more detail or at least present representative pictures from more cells to convince the reader that there is no effect on the apical microtubule cluster.
- The hypothetical model presented is, in my opinion, is the weakest part of the manuscript. I suggest that the authors include the description of this figure in the discussion section, not in the

results. I also encourage the authors to expand the options for possible models, while I understand that there are still many unknowns to the mechanisms of control of tip-growth in plants. Kinesin 14-IIb is a minus-end-directed motor, hence it should be predicted to transport cargo away from the tip, as correctly proposed by the authors. Nevertheless, the default condition (when kinesin 14-IIb is absent) seems to be a positive gravitropic response, suggesting that removing cargo reverses this hypothetical default response. Also, it would be valuable to try to frame their model as part of the vesicle clustering model for polarization and growth proposed by Furt et al. in 2013. -Most of the imaging is of high quality. Nevertheless, it will be valuable to solidify their conclusions by using images of the motors moving on microtubules obtained by dual-color microscopy. In the current version, only static images of the kinesin particles and microtubules are shown. This additional data is important because one possible interpretation of the motility data presented figure 4C is that the kinesin 14-IIb is tracking depolymerizing plus ends. A recently identified molecule from moss (see Ding et al. 2018 doi: 10.1371/journal.pgen.1007221) shows this type of behavior. This should not be difficult, because the authors have the necessary cell lines and expertise to do the dual-color analysis. Furthermore, it will also be valuable to report the average velocity of the kinesin particles to be able to compare their results with existing values for other members of this subfamily.

-It is puzzling that the authors do not cite the work of Ding et al. 2018 doi:10.1371/journal.pgen.1007221, where a microtubule depolymerizing-end-tracking molecule that affects polarized growth was recently identified. In addition, not only is this work relevant because of the microtubule association, but the work of Ding et al. used a similar strategy to the presented in the current manuscript to identify the genetic lesion of the gravitropic mutant. I encourage the authors to discuss the work of Ding et al. concerning the role of microtubules in tip growth and to contrast the genetic mapping strategies.

Minor comments

-In lines 97-98 when the first mention of the kinesin gene subfamily is made in the manuscript, it would be appropriate to cite the original article where the subfamily was first identified (Shen et al. *Frontier Front. Plant Sci.* 2012. doi.org/10.3389/fpls.2012.00230). It would also be valuable to contrast, in the discussion, the phylogenetic analyses that were done in 2012 against the phylogenetic analyses reported in the current manuscript.

-It will be valuable to mention if other reversed response gravitropic mutants that do not correspond to the kinesin 14-IIb gene were identified during the genetic screen.

-The term "higher plants" (line 33) could be replaced with seed plants or a more specific term.

Reviewer #3 (Remarks to the Author):

The manuscript by Yufan Li and collaborators (NCOMMS-20-00005: "Which Way Up is Down? A Molecular Motor Directs Gravitropism in Tip-Growing Plant Cells") describes the isolation and characterization of several mutations that affect protonemal gravitropism in *Physcomitrella patens*. The authors identify several new mutations that result in opposite gravitropism relative to wild type. They show these mutations to be allelic to a previously identified *gtrc* mutation, resulting in similar phenotypes. They clone the corresponding gene using map-based approaches, and show it encodes a minus-end directed kinesin of the KCH (or 14-IIb) family. They demonstrate that the N- and C-terminal ends of the protein, as well as its motor domain, are needed for activity in gravitropism whereas an actin binding domain is not. They also show that this kinesin differs from other members of the same clade in its ability to modulate gravitropism, a specificity more specifically associated with the N terminus of the protein. Importantly, they use functional GFP fusions to demonstrate that this kinesin moves toward the - end of microtubules, away from the tip of the protonemata end cells. This process is critical for proper localization of the actin cluster at the tip of the protonema, which is known to dictate the direction of tip growth. In its presence, gravistimulation leads to actin cluster localization toward the upper end of the protonema, directing upward curvature. In its absence, the cluster moves toward the lower end of the protonema, leading to downward curvature. Hence, movement of an undefined cargo away from the tip on microtubules is needed for proper localization of the actin cluster at the tip, and negative gravitropic curvature.

This well-written manuscript provides exciting new information linking cytoskeleton functions to

single-cell gravitropism is *Physcomitrella*. The genetic analysis involves several independently isolated mutations in the same gene, and functional rescue experiments that demonstrate the authors have isolated the right locus. Utilization of functional GFP-KCH fusions to demonstrate association with the microtubules and mobility toward the – end, away from the tip of the cells, is also quite convincing. Finally, demonstration that the opposite gravitropic curvature displayed by mutant protonemata is associated with early repositioning of the actin cluster toward the bottom rather than top of the cell tip is also exciting. These observations are likely to be of interest to a broad readership. This being said, I have several concerns about this manuscript, which are provided below.

1. While the data clearly show an effect of mutations in this kinesin gene on the positioning of the actin cluster at the tip of gravistimulated protonemata and its impact on the direction of gravitropism, the mechanisms involved in this process remain largely unknown.

2. There is very little attempt to quantify some of the processes under investigation. This is especially true when it comes to analyzing the movement of GFP-marked kinesin along microtubules, and the repositioning of the actin cluster at the tip of graviresponding cells. How many cells were analyzed in these studies? Of those analyzed, what fraction actually gave the patterns described in this manuscript?

3. The authors claim that mutations in this *gtrC* gene do not affect organization of the MT network upon gravistimulation. This information should be documented in Supplementary Materials.

4. The movement of the GTRC kinesin away from the tip toward the minus end of the microtubules is documented with one figure and a movie. Considering the importance of this observation and its potential impact on positioning of the actin cluster at the tip of the cell, it would seem appropriate to better characterize this movement in both unstimulated and also gravistimulated protonemata. An important question that comes to mind in view of the observed data and the model proposed in this manuscript, is whether this movement remains symmetrical in gravistimulated protonemata? Is it possible that the movement itself is affected on one side of gravistimulated cells (top or bottom)? If this were the case, the cargo, which appears to play a key role in defining the polarity of actin cluster relocalization and curvature response, could be viewed as a key determinant of response polarity. Therefore, it would seem appropriate to expect a spatio-temporal analysis of kinesin movement away from the tip between upper and lower halves of gravistimulated cells (relative to controls).

5. A few additional minor comments/suggestions follow:

a. Abstract, line 24. It seems that GTRC is assigned the name KCHa instead of KCHb.

b. Page 6, lines 130-134, and Figure 4B. Considering that only one concentration of latrunculin B and oryzalin are used in the experiment testing the need for actin and microtubules for proper localization of GTRC, a control should be included in Supplementary Data showing that these treatments truly affect their respective cytoskeleton components under the conditions tested.

c. The kinetics of gravitropic response presented in figure 6b are different between wild type and mutant (faster response in the mutant relative to wild type). This point should be discussed in the manuscript.

d. Page 8, line 196. Please specify which mutant(s) was checked for actin distribution upon gravistimulation.

e. In Methods, the description of plasmid constructions is very lengthy and rather tedious to read. It should probably be moved to Supplementary Data.

f. In Figure 2, a and b, the left panels are rather confusing. It's difficult to deduce what is what. The genotype name and segregant numbers are misaligned with the colonies. Furthermore, the direction of protonemata growth seems to not align with the names given at the top (up; down; etc). Finally, the rows are not labeled. Are we simply looking at a number of segregating progenies, lined up along each row, without association with the phenotypes described on top of the figure? These panels should be better explained in the legend.

g. In figure 4, the kymograph would be easier to interpret if the time was indicated along the X axis and the position of the kinesin along the microtubules were indicated along the Y axis (this would be compatible with the first three figures of this panel, facilitating interpretation of the results).

h. As specified earlier, the numbers of cells analyzed for fluorescence signals should be specified, as should the proportion that gave different results relative to the shown data. Also, how many times was each experiment repeated?

Response to the review comments

We appreciate these review comments, which are extremely valuable! We have successfully completed all the essential experiments and revised the manuscript substantially. The changes in the manuscript are highlighted in Yellow. Our point-by-point responses to the review comments are listed below.

Reviewer #1 (Remarks to the Author):

The major claim by Li et al is the identification of a gene for required for defining the direction (up versus down) of gravity response. When mutated the loss of this gene function leads to a reversed direction of protonemal gravitropism in the *gtrC* mutant. The authors claim that this gene is a KCHb that was previously reported as a member of kinesin-14 regulating nuclear transport and cytoskeletal coalescence for tip growth. They indicate that phenotype associates with a defined lesion and confirm this by showing that the phenotype can be repaired by introducing a wild type copy of the gene by transformation. They further confirm gene identity by KO of the candidate by 2 independent means. This aspect of the work seems very secure. Using a variety of other defined mutations and inferring from what is known about the kinesin gene family, the authors argue that the motor domain is necessary for function and show that the GFP fusion protein localises in a MT dependent manner. All these findings are consistent with its being a functional kinesin.

The manuscript is well organized and the finding is interesting in the research field on gravity sensing/response since this work reveals a mechanism whereby these cells can translate a directional signal into the growth response of the correct direction. Therefore, it should be of interest to quite a wide readership. There are some concerns about some of the detailed mechanistic conclusions, which should be addressed:

Comment 1: Based on the apparent lack of MT localisation of the truncated KCHb fused to GFP, the C terminus is proposed to be essential for MT localisation. Indeed, it seems that no KCHb dependent GFP signals are detected in fig.5b. However, is the truncation of the C terminus is likely to lead to protein instability and thereby rapid degradation, they should check the level of the GFP-fused protein (e.g. WB with anti-GFP antibody).

Response: Following the suggestion, we used anti-GFP antibody to check the level of the GFP-

fused proteins via Western blot (Supplementary Fig. 8). Although N terminally truncated GTRC was not detected in our Western blot assay probably due to its special protein properties, the fluorescence has clearly showed its microtubule localization in tip cells (Fig. 5b). All the other protein forms have been detected successfully. The protein levels of C terminus truncated and G856R forms were higher or comparable to that of full-length GTRC, indicating that the deletion of C terminus or mutation in motor domain affect GTRC's microtubule-binding activity rather than its protein stability.

Comment 2: It is claimed that KCHb has a unique function in gravity signaling because no roles of both KCH a and c were not detected. How about KCHd? What are the gravi-phenotypes of double *a/c* or other combinations. These should be easily checked as all four of members of the family have been KOed?

Response: We generated several additional mutants to further support the unique role of KCHb in gravity signaling (Fig. 3g). For all the single mutants, *gtrc/kchb* is the only one showing reversed gravitropic phenotype, while *kcha*, *kchc* and *kchd* did not show any obvious phenotype. Further, neither did the double mutant *kcha kchc* or the triple mutant *kcha kchc kchd* show an obvious gravitropic phenotype. The quadruple mutant *kcha kchb kchc kchd* showed the reversed gravitropic phenotype, as well as serious growth defects. Together, the redundant roles of *KCH* subfamily genes in the growth revealed by the quadruple mutant were consistent with previous study (Yamada et al., 2018, Plant Cell), whereas the specific function of KCHb in gravity signaling is unique.

Comment 3: KCHa did not complement the mutant phenotype so the authors claim that the biochemical activity of KCHb is different from those of KCHa. This claim is weak since no direct biochemical data is presented in the manuscript. The complementation test basically provides the biological property of the kinesin gene, not directly its biochemistry and there are many other reasons for lack of complementation.

Response: Thanks for the suggestion. We used “biological property” to replace the “biochemical activity” in the text.

Comment 4: It is proposed that the predominant function of the KCHb's N-terminal region is in

gravity sensing. But the C-terminal region, as well as the N-terminal region, is essential for sensing and responding to gravity (fig.5a, b). The reciprocal domain swapping test between KCHb and KCHa suggest that the N-terminal region of KCHb is more divergent from that of KCHa, but again protein levels need to be checked to eliminate the more obvious confounding factors (such as protein instability).

Response: Since no tag was fused with the domain-swapped proteins, we could not compare the protein levels directly. As a proxy, we have used RT-PCR to examine the transcripts of *KCHa*, *KCHb* and the two chimeric *KCHs*, and found that their transcriptional levels were comparable (Supplementary Fig. 9), indicating that these genes are expressed to the same extent.

Comment 5: The importance of an apically located actin cluster is suggested but the supporting evidence is quite weak and lacks sufficient quantification. The actin cluster accumulates at the upper region of the apical dome of wild type in response to the direction of gravity in wild type, while the MT foci did not. It is a bit curious because the AF cluster co-localises to the MT foci in the expanding apical dome under normal growth condition. This implies that, during the gravitropic growth response, the cytoskeletal organisation, or dynamics, within the apical dome is different from normal growth? If so, the effect of gravity on MT distribution and the MT dynamics both need quantification - are they altered in the apical dome and does the mutation affect either? In the Fig.6e schematics, the MT foci preferentially accumulate at the upper part of the apical dome, together with the AF cluster. This should be easy to clarify.

Response: To further characterize the apically located actin cluster, we added quantification to compare the actin accumulation in the upper and lower regions of the apical dome in the tip cells after gravistimulation (Supplementary Fig. 10). Tip cells that keep growing horizontally were excluded from the quantitative analysis because they might represent the cells losing gravitropic response due to laser effect. For the protonemata showing a gravitropic response, 100% (10/10) wild type protonemata showed upward gravitropic bending after horizontally placement, and actin clusters were always accumulated in the upper region of these protonema tip cells. For the *gtrC* mutant, although eventually all protonemata grew downwards as shown in Figures 1 and 6, the protonemata were wavy and showed different directions during the growth. For the first obvious bending of *gtrC* after gravistimulation, 70% (7/10) protonema showed downward bending and 30% (3/10) showed upward bending, in which actin clusters were always accumulated in the side of

bending direction. Together, the asymmetric distributions of actin clusters are obviously different in *gtrC* mutant and wild type.

Considering microtubules, although the growth of TUB-GFP/*gtrC* is seriously retarded, our new statistical analysis using few short protonemata showed that mutation of *GTRC* had no obvious effect on microtubule orientations under regular growth conditions (Supplementary Figure 11). Under the gravistimulation condition, for those TUB-GFP protonemata showing negative gravitropic responses, the MT foci were clearly located on the upper side of apical region ahead of bending (Supplementary Figure 12). Since almost no growth could be detected during several - hours observation, the few protonemata of TUB-GFP/*gtrC* could not be used to monitor the gravity-triggered MT orientation, whereas our expectation is that the MT foci in the *gtrC* mutant should locate on the lower side of apical region ahead of bending. Generally, the MT foci should co-localize with the AF cluster in the expanding apical dome, under both normal growth (Wu and Bezanilla, 2018, J Cell Biol), and gravistimulation conditions as shown in the schematics in Fig. 6e, which needs to be further clarified in the future.

We also generated another line: p7113-mRFP-tub for studying the effect of gravity on microtubule distribution and dynamics. However, this line showed a strong signal in the basal parts of the dark-grown protonema but a very faint signal in the tip cells (Response Fig. 1). We have to use a high-power laser and quite long exposure time to collect the signal (Fig. 4a), and the long-time and dynamic observation is impossible.

Response Figure 1. Microtubule signal intensity in the protonema transformed with p7113-mRFP-tub. The RFP fluorescence in the dark-grown protonema was collected by a ZEISS LSM800 confocal microscope. The corresponding bright field view was shown on the right. The arrow labeled with “g” indicates the direction of gravity. Scale bar, 5 μ m.

Comment 6: the lack of role in nuclear positioning is weakly supported, with only a pair of images (Supplementary figure 6 - the mutant nucleus seems displaced towards the tip but its significance is impossible to assess). L222 and 223 refers to evidence for this in Fig5 and supp fig 6 but i dont see any in Fig 5? Quantitative data is required to support this nuclear position claim, and ideally also from double and triple mutants if available.

Response: Following the suggestion, we have now analysed the the nuclear positioning more rigorously (Supplementary Fig. 4). The analyses of *gtrC/kchb*, *kcha kchc*, *kcha kchc kchd* and *kcha kchc kchb kchd* mutants clearly showed that GTRC/KCHb itself does not affect the nuclear positioning, while the KCH subfamily members control the nuclear position redundantly. This result is consistent with a previous report (Yamada et al., 2018, Plant Cell). Combining this with the phenotype of *kch* mutants (Fig. 3g), we concluded that there is no direct correlation between

nuclear position and the gravity response, and that the role of GTRC/KCHb in controlling the gravity response is unique among the *KCH* subfamily genes.

Comment 7: Other minor points include:

1. P6, L135-136. The minus-end-directed movement should be assessed with MT dynamics that is associated with KCHb .

Response: Following this suggestion, we recorded the movement of KCHb on the MT using the GFP-GTRC/RFP-TUB transgenic line (Supplementary Video 2). Since the RFP-TUB signal is too weak in the dark-grown protonema tip cells, we used the light-grown protonema in this experiment.

2. P10, L255. It should be added in the “Plant materials and growth conditions” section how gravitropic assay is carried out (e.g. medium and cultivation condition). BCD medium for this assay is correct? In dark condition, the medium is sometime supplemented with glucose or sucrose as a carbon source.

Response: To carry out the gravitropic assay, *P. patens* was cultured on BCD or BCDAT medium with 1% (w/v) sucrose and 0.5-1% (w/v) agar at 25 °C under continuous white light for around a week, and then the culture plates were kept vertically in the dark for the indicated times as described in the text. For gravistimulation, these plates were further rotated for 90 degrees. These information about gravitropic assay has been added into the revised manuscript.

3. P10, L261. “50mM” may be replaced with “50 mM”.

4. P13, L347. “Xba I” may be replaced with “Xba I”.

5. P13, L347. “PTN182” may be replaced with “pTN182”.

6. P13, L356. “1537bp” may be replaced with “1537-bp”.

7. P14, L365. “1537bp” may be replaced with “1537-bp”.

8. P15, L413. “100mM” may be replaced with “100 mM”.

Response to 3-8: We have revised the manuscript following these suggestions.

9. P15, L417. The preparation of the cells for observation is described in the “Microscopic observation” section. E.g. medium? What type of glass-bottom dish type was used?

Response: The preparation of the samples, including medium and the glass-bottom dish type, have

been added into the “Microscopic observation” section. For preparation of the cells for observation, 2 ml BCDAT medium with 1% (w/v) agar and 1% (w/v) sucrose was added into a glass-bottom dish (Nest Scientific 801001). After the medium solidified, a central core was excised and replaced by a thin layer of the same medium just covering the glass base.

10. P25, fig.3d. unfortunately, the southern blot is confusing. Please provide more information on this blot.

Response: Transformation in moss sometimes causes multi-site insertions that can confound phenotypic analysis. To eliminate this possibility in our study, a Southern blot assay was performed to confirm that no off-target integration occurred. This information has been added into our revised manuscript, with details in the figure legend and methods.

11. P35, L704. “Supplementary Fig. 1d” is incorrect? Please mention the fig that shows the primer position.

Response: Sorry about this error. The figure should be Fig. 2c, which has been corrected in the revised manuscript.

12. P37, L720 and 723 (supplemental fig.4 b and d). These panels look agarose-gel electrophoresis..., perhaps it's not DNA-gel blot analysis.

Response: Thanks for pointing this out. It should be agarose-gel electrophoresis, which have been corrected in the revised manuscript.

Reviewer #2 (Remarks to the Author):

The manuscript by Li et al. presents a thorough characterization of a reversed response gravitropic mutant of *Physcomitrella patens*. Although mutants that have this reversed response were initially isolated more than 30 years ago, the identity of the genetic lesion was unknown until now. The fact that the mutated gene is a kinesin that belongs to a subfamily with four members is surprising and reveals an exciting type of subspecialization of kinesin function. The observation that the apical actin cluster is positioned at the opposite side in the mutants is puzzling, and understanding the mechanism of this will prove to be very important for our future understanding of signaling

directed to the tip-growth machinery.

Major comments:

Comment 1: The title, as it is, is confusing and could be more appropriate for a review article. I suggest crafting a title so that it clearly states the content and conclusions of the manuscript.

Response: Thanks for the suggestion. The title has been revised to be “A minus-end directed kinesin motor directs gravitropism in *Physcomitrella patens*”.

Comment 2: The descriptions for the mapping of the mutation are somewhat confusing. For example, it is not clear how the spores resulting from the outcross were selected from the ones resulting from selfing. It will also be useful to compare the two mapping approaches used for efficiency, cost, complexity, and time required.

Response: Following this suggestion, we have added the descriptions about how the spores resulting from the outcross were selected in the revised manuscript. The Gransden (Gd) wild type that we used for the mutant screening showed reduced male-fertility, and generated sporophytes by selfing slowly. Thus, the spores we collected from the majority of sporophytes growing on the Gransden parental strain were outcrossed, which was further confirmed by segregation analysis and genotyping.

The comparison of the two mapping approaches used in our study has been added into the revised manuscript in the Discussion section. The traditional mapping strategy we used for *gtrC-5* mapping used an initial population of 20 segregants, genotyped in the SNP mapping process to generate the chromosome-scale genome assembly. Subsequently a population of 282 segregants were genotyped for a small number of SNPs and SSR within the genetic interval identified. Then, candidate genes within the interval were sequenced. The high-throughput sequencing strategy we carried out on *gtrC-16* required two runs of high-throughput sequencing: one for the mutant itself and one for pooled segregants. Around 30 segregants were enough to generate a high-resolution SNP ratio map and no further sequencing was required. The advantage of traditional mapping strategy is simple analysis, while the principal disadvantage is that it is labor- and time-consuming in genotyping and candidate gene sequencing. The advantage of high-throughput sequencing strategy is its speed and efficiency, whereas its complexity is requirement for bioinformatic analysis capability. The costs are comparable.

Comment 3: The lack of a nuclear positioning phenotype is not surprising, but the analysis, as presented, is not thorough. Only one cell is shown in supplementary Fig. 6, and no quantification of nuclear positions that could rule out a partial phenotype is present for the mutant.

Response: Thanks for the suggestion. We have added the quantification of the nuclear positions of the *gtrC/kchb* mutant and wild type. Several newly generated *kch* mutants were also included for comparison (Supplementary Fig. 4 in the revised manuscript).

Comment 4: Similarly to above, the lack of MT organization phenotype at the tip is not thoroughly characterized. Only one image is presented as supplemental figure. This is a good quality image, but because microtubules have been shown to help position the apical actin cluster, it will be helpful to characterize the microtubule organization at the tip in more detail or at least present representative pictures from more cells to convince the reader that there is no effect on the apical microtubule cluster.

Response: Following the suggestion, we have added additional representative pictures of MT organization from different cell tips. Also, we have quantified the microtubular orientation in these cell tips (Supplementary Fig. 11). In addition, we carried out gravistimulation using a TUB-GFP line. For those TUB-GFP protonemata showing negative gravitropic responses, the MT foci were clearly located on the upper side of apical region ahead of bending (Supplementary Figure 12).

Comment 5: The hypothetical model presented is, in my opinion, is the weakest part of the manuscript. I suggest that the authors include the description of this figure in the discussion section, not in the results. I also encourage the authors to expand the options for possible models, while I understand that there are still many unknowns to the mechanisms of control of tip-growth in plants. Kinesin 14-IIb is a minus-end-directed motor, hence it should be predicted to transport cargo away from the tip, as correctly proposed by the authors. Nevertheless, the default condition (when kinesin 14-IIb is absent) seems to be a positive gravitropic response, suggesting that removing cargo reverses this hypothetical default response. Also, it would be valuable to try to frame their model as part of the vesicle clustering model for polarization and growth proposed by Furt et al. in 2013.

Response: Following the suggestion, we have moved the description of this hypothetical model

into the discussion section. Also, we tried to expand the discussion about the possible model by integrating the results of our and previous studies, including Furt et al., 2013.

Comment 6: Most of the imaging is of high quality. Nevertheless, it will be valuable to solidify their conclusions by using images of the motors moving on microtubules obtained by dual-color microscopy. In the current version, only static images of the kinesin particles and microtubules are shown. This additional data is important because one possible interpretation of the motility data presented figure 4C is that the kinesin 14-IIb is tracking depolymerizing plus ends. A recently identified molecule from moss (see Ding et al. 2018 doi: 10.1371/journal.pgen.1007221) shows this type of behavior. This should not be difficult, because the authors have the necessary cell lines and expertise to do the dual-color analysis. Furthermore, it will also be valuable to report the average velocity of the kinesin particles to be able to compare their results with existing values for other members of this subfamily.

Response: This is a good suggestion. We have recorded the movement of GTRC on microtubules by dual-color microscopy (Supplementary video 2). In our GFP-GTRC/RFP-TUB transgenic line, the RFP-TUB signal is very weak in the tip cell, but the result clearly showed that GTRC/KCHb is a minus-end-directed kinesin. This is consistent with a previous study that used KCHa as an example to prove that KCHs are minus-end directed motors (Yamada et al., 2018, Plant Cell). Thus, the behavior of GTRC/KCHb is distinctly different compared to those proteins tracking depolymerizing plus ends as reported in Ding et al. 2018, PLoS Genetics.

We have also calculated the average movement velocity of GTRC/KCHb, and the data have been added into the revised manuscript (Fig. 4d). The speed of KCHb (463 ± 122 nm/s) calculated in our study is very similar the reported speed of KCHa (441 ± 226 nm/s) (Yamada et al., 2018, Plant Cell).

Comment 7: It is puzzling that the authors do not cite the work of Ding et al. 2018 doi:10.1371/journal.pgen.1007221, where a microtubule depolymerizing-end-tracking molecule that affects polarized growth was recently identified. In addition, not only is this work relevant because of the microtubule association, but the work of Ding et al. used a similar strategy to the presented in the current manuscript to identify the genetic lesion of the gravitropic mutant. I encourage the authors to discuss the work of Ding et al. concerning the role of microtubules in tip

growth and to contrast the genetic mapping strategies.

Response: Thanks for bringing this to our attention. We have now cited this interesting work in our revised manuscript, to discuss both of the important roles of microtubules in tip growth and to reinforce the significant potential of the forward genetic mapping strategies that are now being used in the model *P. patens* system.

Comment 8: Minor comments

1, In lines 97-98 when the first mention of the kinesin gene subfamily is made in the manuscript, it would be appropriate to cite the original article where the subfamily was first identified (Shen et al. *Frontier Front. Plant Sci.* 2012. doi.org/10.3389/fpls.2012.00230). It would also be valuable to contrast, in the discussion, the phylogenetic analyses that were done in 2012 against the phylogenetic analyses reported in the current manuscript.

Response: Following this suggestion, we have added the citation of Shen et al. 2012 in our revised manuscript. Also, the comparison of the phylogenetic analyses between Shen *et al.* and our own analysis has been discussed in the results.

2, It will be valuable to mention if other reversed response gravitropic mutants that do not correspond to the kinesin 14-IIb gene were identified during the genetic screen.

Response: We have obtained another mutant showing a reversed gravitropic response and the corresponding gene is not kinesin 14-IIb. We have mentioned this in the revised manuscript. The analysis of this second gene is proceeding and will be published once it has been fully characterized.

3, The term "higher plants" (line 33) could be replaced with seed plants or a more specific term.

Response: The term "higher plants" has been replaced with "vascular plants" in the revised manuscript.

Reviewer #3 (Remarks to the Author):

The manuscript by Yufan Li and collaborators (NCOMMS-20-00005: "Which Way Up is Down? A Molecular Motor Directs Gravitropism in Tip-Growing Plant Cells") describes the isolation and

characterization of several mutations that affect protonemal gravitropism in *Physcomitrella patens*. The authors identify several new mutations that result in opposite gravitropism relative to wild type. They show these mutations to be allelic to a previously identified *gtrc* mutation, resulting in similar phenotypes. They clone the corresponding gene using map-based approaches, and show it encodes a minus-end directed kinesin of the KCH (or 14-IIb) family. They demonstrate that the N- and C-terminal ends of the protein, as well as its motor domain, are needed for activity in gravitropism whereas an actin binding domain is not. They also show that this kinesin differs from other members of the same clade in its ability to modulate gravitropism, a specificity more specifically associated with the N-terminus of the protein. Importantly, they use functional GFP fusions to demonstrate that this kinesin moves toward the – end of microtubules, away from the tip of the protonemata end cells. This process is critical for proper localization of the actin cluster at the tip of the protonema, which is known to dictate the direction of tip growth. In its presence, gravistimulation leads to actin cluster localization toward the upper end of the protonema, directing upward curvature. In its absence, the cluster moves toward the lower end of the protonema, leading to downward curvature. Hence, movement of an undefined cargo away from the tip on microtubules is needed for proper localization of the actin cluster at the tip, and negative gravitropic curvature.

This well-written manuscript provides exciting new information linking cytoskeleton functions to single-cell gravitropism in *Physcomitrella*. The genetic analysis involves several independently isolated mutations in the same gene, and functional rescue experiments that demonstrate the authors have isolated the right locus. Utilization of functional GFP-KCH fusions to demonstrate association with the microtubules and mobility toward the end, away from the tip of the cells, is also quite convincing. Finally, demonstration that the opposite gravitropic curvature displayed by mutant protonemata is associated with early repositioning of the actin cluster toward the bottom rather than top of the cell tip is also exciting. These observations are likely to be of interest to a broad readership. This being said, I have several concerns about this manuscript, which are provided below.

Comment 1: While the data clearly show an effect of mutations in this kinesin gene on the positioning of the actin cluster at the tip of gravistimulated protonemata and its impact on the direction of gravitropism, the mechanisms involved in this process remain largely unknown.

Response: It is true that the mechanism about how the kinesin controls the actin cluster at the tip

of gravistimulated protonemata has not been revealed yet. We have added further discussion about the possible events involved in this process, including the redistribution of microtubule foci and myosin XI-associated structures. We are trying to identify more factors involved in this regulatory process, which should reveal more underlying mechanisms but it takes time.

Comment 2: There is very little attempt to quantify some of the processes under investigation. This is especially true when it comes to analyzing the movement of GFP-marked kinesin along microtubules, and the repositioning of the actin cluster at the tip of graviresponding cells. How many cells were analyzed in these studies? Of those analyzed, what fraction actually gave the patterns described in this manuscript?

Response: Following this valuable suggestion, we have thoroughly quantified these parameters. For GFP-marked kinesin movement, the velocity distribution was analyzed and added as Fig. 4d. The repositioning of the actin cluster was quantitatively analyzed, and added as Supplementary Fig. 10. In these analyses, the cell numbers were clearly indicated and the fractional information was provided. All the source data were included as well.

Comment 3: The authors claim that mutations in this *gtrC* gene do not affect organization of the MT network upon gravistimulation. This information should be documented in Supplementary Materials.

Response: Our observations indicate that the general organization of the MT network appears unaffected in *gtrC* knock-out mutant under regular growth condition. We have added more images and further quantification in Supplementary Fig. 11. In addition, we carried out gravistimulation using a TUB-GFP line. For those TUB-GFP protonemata showing negative gravitropic responses, the MT foci were clearly located on the upper side of apical region ahead of bending (Supplementary Figure 12). Since the TUB-GFP/*gtrC* line developed very short protonemata in darkness (a priority for observing gravitropic responses) and didn't show obvious growth within several hours, thus it was difficult for us to carry out gravistimulation analysis (Supplementary Fig. 11). Therefore, we could only speculate that the MT foci in the *gtrC* mutant locate on the lower side of apical region ahead of bending, and further study will be needed to clarify this question.

Comment 4: The movement of the GTRC kinesin away from the tip toward the minus end of the

microtubules is documented with one figure and a movie. Considering the importance of this observation and its potential impact on positioning of the actin cluster at the tip of the cell, it would seem appropriate to better characterize this movement in both unstimulated and also gravistimulated protonemata. An important question that comes to mind in view of the observed data and the model proposed in this manuscript, is whether this movement remains symmetrical in gravistimulated protonemata? Is it possible that the movement itself is affected on one side of gravistimulated cells (top or bottom)? If this were the case, the cargo, which appears to play a key role in defining the polarity of actin cluster relocalization and curvature response, could be viewed as a key determinant of response polarity. Therefore, it would seem appropriate to expect a spatio-temporal analysis of kinesin movement away from the tip between upper and lower halves of gravistimulated cells (relative to controls).

Response: Thanks for this suggestion. Actually, we also hoped to find some asymmetric pattern of GTRC movement between upper and lower halves of gravistimulated cells. We spent a lot of time observing the fluorescence of GFP-GTRC or mCherry-GTRC under gravistimulation using the ZEISS LSM800 or LSM710 laser confocal microscopes with an adjusted vertical objective table, and specifically searched for a redistributed polarity of GTRC. However, to date, we have not detected any obvious asymmetric distribution of GTRC during gravistimulation (Response Fig. 2). However, we cannot completely exclude the possibility of asymmetrical movement of GTRC proteins under gravistimulation. This will require further analysis using advanced microscopy and analysis strategies in the future, such as setting up a spinning disk confocal microscopy system with a vertical objective table, which needs an extended period.

Response Figure 2. The distribution of GFP-GTRC or mCherry-GTRC during gravistimulation. **a, b,** Protonemata expressing GFP-GTRC were grown in the dark vertically for 1 week and then turned 90 degree to be gravistimulated, and the fluorescence was collected by a ZEISS 800 confocal microscope. No obvious redistribution of GFP-GTRC signal in our lines was observed at 30 minutes (**b**) compared to 0 minute (**a**). Three representative tip cells were shown for each time point. Arrows labeled with “g” indicate the direction of gravity vectors. **c, d,** Two mCherry-GTRC tip cells were grown in the dark vertically for 1 week and then turned 90 degree to be gravistimulated, and the fluorescence was collected by a ZEISS 710 confocal microscope at the indicated time points. The relative intensity of mCherry-GTRC is shown as colors using a “Green fire blue” look-up table where the high intensity is represented by green (maximum white) and the low intensity is represented by blue. Scale bars, 10 μ m

Comment 5: A few additional minor comments/suggestions follow:

a. Abstract, line 24. It seems that GTRC is assigned the name KCHa instead of KCHb.

Response: This mistake has been corrected in the revised manuscript.

b. Page 6, lines 130-134, and Figure 4B. Considering that only one concentration of latrunculin B and oryzalin are used in the experiment testing the need for actin and microtubules for proper localization of GTRC, a control should be included in Supplementary Data showing that these treatments truly affect their respective cytoskeleton components under the conditions tested.

Response: Thanks for the suggestion. In the Supplementary Fig. 7, we have added the treatments of oryzalin and latrunculin B in relation to microtubules and actin respectively, as the controls for the studies in Fig. 4 b.

c. The kinetics of gravitropic response presented in figure 6b are different between wild type and mutant (faster response in the mutant relative to wild type). This point should be discussed in the manuscript.

Response: We have added some discussion of the kinetic differences between wild type and *gtrC* mutants in the revised manuscript. We suggest that the direction of gravitropism is controlled by two groups of factors. One group promotes the upward growth of protonemata, whereas another group promotes the downward growth. GTRC is one of the factors maintaining the upward growth of protonemata. The kinetics of gravitropic response depend on the competence of these two groups of factors

d. Page 8, line 196. Please specify which mutant(s) was checked for actin distribution upon gravistimulation.

Response: We have specified this mutant in the revised manuscript, in both of the Result and Method sections. GTRC-HRKO plasmid was transformed into a *Lifeact-mCherry* line to generate the *Lifeact-mCherry/gtrC* line.

e. In Methods, the description of plasmid constructions is very lengthy and rather tedious to read. It should probably be moved to Supplementary Data.

Response: Following this suggestion, we have kept only the core steps of plasmid construction in the main text, and moved the details of plasmid construction into the supplementary data.

f. In Figure 2, a and b, the left panels are rather confusing. It's difficult to deduce what is what. The genotype name and segregant numbers are misaligned with the colonies. Furthermore, the direction of protonemata growth seems to not align with the names given at the top (up; down; etc). Finally, the rows are not labeled. Are we simply looking at a number of segregating progenies, lined up along each row, without association with the phenotypes described on top of the figure? These panels should be better explained in the legend.

Response: To avoid confusion, we have revised these panels to be two separated parts (Figure 2a and 2b). The numbers (No.) of each phenotype were shown in the tables below each image. The original figures included both of the representative segregating progenies and the statistic numbers of the phenotypes, but there's no alignment information between them.

g. In figure 4, the kymograph would be easier to interpret if the time was indicated along the X axis and the position of the kinesin along the microtubules were indicated along the Y axis (this would be compatible with the first three figures of this panel, facilitating interpretation of the results).

Response: Following the suggestion, we have adjusted Fig. 4c, using X axis to indicate the time and Y axis to indicate the position.

h. As specified earlier, the numbers of cells analyzed for fluorescence signals should be specified, as should the proportion that gave different results relative to the shown data. Also, how many times was each experiment repeated?

Response: Following the suggestion, the quantitative analysis information, including the numbers of cells analyzed for fluorescence, the proportion that gave different results, and number of experimental repeats have been included in the revised manuscript.

REVIEWER COMMENTS

Reviewer #2 (Remarks to the Author):

The authors addressed all of my comments successfully. The research presented is of high quality and will be of much interest to the cell biology community, in particular those interested in the cytoskeleton, intracellular transport, and gravity perception.

Reviewer #3 (Remarks to the Author):

This revised version of the manuscript by Yufan Li et al. (Title: "A minus-end directed kinesin motor directs gravitropism in *Physcomitrella patens*") demonstrates a role for a minus-end directed kinesin of the KCH type in protonema gravitropism in *Physcomitrella*. As mentioned in my review of the previous draft, this well-written manuscript provides exciting new information about the molecular mechanisms that govern single-cell gravitropism in this system. It should be of interest to a broad readership. In this revised draft, the authors have addressed most of the comments and suggestions raised by all three reviewers, thereby dramatically improving the quality of their manuscript. This being said, the paper still lacks a clear mechanistic explanation of the change in tip growth direction upon gravistimulation displayed by *gtrc* mutants relative to wild type, leading to a rather long model in the discussion attempting to reconcile the various pieces of data reported in this work. While the paper still provides important novel information on the process (involvement of a specific kinesin contributing to asymmetric tip-localization of the actin bundle upon gravistimulation), the basic mechanism of growth polarity change upon gravistimulation remains unexplained, which seems rather unsatisfying for a journal of Nature Communications stature.

Apart from this general comment, I also have a couple of additional minor suggestions:

- 1) In Figure 4, the described colocalization between GFP-GTRC and MT signals should be quantified.
- 2) In the legend to Supplementary Figure 11, the authors claim that the histograms represented in panels b and c represent "numbers of (MT) structures in given orientations". However, the Y axis shows decimal numbers, suggesting these histograms actually show fractions of MT in defined orientations. If this is true, please indicate the total numbers of MTs analyzed for each genotype.
- 3) In the experiment testing the effect of domain swapping between KCHa and KCHb on gravitropism rescue, using RT-PCR to verify expression similarity between constructs is suboptimal as it really tells nothing about artifacts potentially associated with differential protein stability.

Response to the review comments

We appreciate all the review comments, and have successfully completed all the required experiments and analyses. The revised or added sentences in the manuscript are highlighted in Yellow. Our point-by-point responses to the review comments are listed below.

Reviewer #2 (Remarks to the Author):

The authors addressed all of my comments successfully. The research presented is of high quality and will be of much interest to the cell biology community, in particular those interested in the cytoskeleton, intracellular transport, and gravity perception.

Response: We appreciate this comment, and also believe this work will attract broad interests.

Reviewer #3 (Remarks to the Author):

This revised version of the manuscript by Yufan Li et al. (Title: “A minus-end directed kinesin motor directs gravitropism in *Physcomitrella patens*”) demonstrates a role for a minus-end directed kinesin of the KCH type in protonema gravitropism in *Physcomitrella*. As mentioned in my review of the previous draft, this well-written manuscript provides exciting new information about the molecular mechanisms that govern single-cell gravitropism in this system. It should be of interest to a broad readership. In this revised draft, the authors have addressed most of the comments and suggestions raised by all three reviewers, thereby dramatically improving the quality of their manuscript. This being said, the paper still lacks a clear mechanistic explanation of the change in tip growth direction upon gravistimulation displayed by *gtrc* mutants relative to wild type, leading to a rather long model in the discussion attempting to reconcile the various pieces of data reported in this work. While the paper still provides important novel information on the process (involvement of a specific kinesin contributing to asymmetric tip- localization of the actin bundle upon gravistimulation),

the basic mechanism of growth polarity change upon gravistimulation remains unexplained, which seems rather unsatisfying for a journal of Nature Communications stature.

Response: Thanks for giving our manuscript the nice appraisements, such as “this well-written manuscript provides exciting new information about the molecular mechanisms that govern single-cell gravitropism in this system”, and “It should be of interest to a broad readership”. Definitely we are very interested in how the kinesin GTRC regulates the growth polarity upon gravistimulation, but it takes time to study and reveal the basic mechanisms. The significance of our current work is identifying a specific kinesin determining the direction of gravitropism, which opens the door for further mechanistic analysis of gravitropism and polar growth.

Apart from this general comment, I also have a couple of additional minor suggestions:

Comment 1: In Figure 4, the described colocalization between GFP-GTRC and MT signals should be quantified.

Response: For quantifying the colocalization between GFP-GTRC and MT signals, we have added scatterplots showing the Pearson’s correlation coefficient (r) between the two fluorescent signals in Figure 4a. The details of this analysis have been added to the Methods.

Comment 2: In the legend to Supplementary Figure 11, the authors claim that the histograms represented in panels b and c represent “numbers of (MT) structures in given orientations”. However, the Y axis shows decimal numbers, suggesting these histograms actually show fractions of MT in defined orientations. If this is true, please indicate the total numbers of MTs analyzed for each genotype.

Response: Thanks for the suggestion. We have altered the term “numbers” to “relative frequency” in the figure legend to avoid confusion. Due to that the software “ImageJ plugin Directionality” can not provide the exact numbers of the structures (MT), we added the estimated total number range of microtubules used for statistical analysis in

these tip cells.

Comment 3: In the experiment testing the effect of domain swapping between KCHa and KCHb on gravitropism rescue, using RT-PCR to verify expression similarity between constructs is suboptimal as it really tells nothing about artifacts potentially associated with differential protein stability.

Response: For demonstrating the proper expression of the chimeric proteins, we re-constructed *GTRC/KCHb*, *KCHa*, and chimeric *KCHs* fused with N-terminal 3×FLAG, and transformed them into *gtrC-16* mutant (Fig. 5c; Supplementary Fig. 6c, d). The phenotypes of these transgenic lines are identical to those without FLAG tags presented in the previous version of our manuscript (Fig. 5d). Western blot analysis clearly showed that the protein levels of GTRC, KCHa and chimeric KCHs are comparable (Supplementary Fig. 9). These results clearly proved the unique role of the N-terminus of GTRC/KCHb in gravitropism.